# Multi-centennial evolution of the climate response and deep ocean heat uptake in a set of abrupt stabilization scenarios with EC-Earth3

Federico Fabiano[1], Paolo Davini[2], Virna L. Meccia[1], Giuseppe Zappa[1], Alessio Bellucci[1], Valerio Lembo[1], Katinka Bellomo[2, 3], and Susanna Corti[1]

[1]Institute of Atmospheric Sciences and Climate (CNR-ISAC), Bologna (Italia)

[2]Institute of Atmospheric Sciences and Climate (CNR-ISAC), Torino (Italia)

[3]Polytechnic University of Turin, Department of Environment, Land and Infrastructure Engineering, Torino (Italia)

**Correspondence:** Federico Fabiano (f.fabiano@isac.cnr.it)

**Abstract.** Understanding long-term committed climate change due to anthropogenic forcing is key to inform climate policies, yet these timescales are still under-explored. We present here a set of 1000-year long *abrupt stabilization* simulations performed with EC-Earth3. Each simulation follows a sudden stabilization of the external forcing at the level specified by CMIP6 for historical (1990) or SSP5-8.5 scenario (2025, 2050, 2065, 2080, 2100) conditions, with a final temperature increase ranging between 1.4 and 9.6 K with respect to the pre-industrial baseline. Remarkably, the simulation stabilized at a GHG level close to present-day (2025) well exceeds in the long term the Paris agreement goals of 1.5 and 2 degrees warming above pre-industrial, and only the 1990 simulation leads to a stabilized climate below 1.5 degrees warming. We first focus on the evolution of the climate response at multi-centennial timescales and its dependence on the level of forcing. We note a decrease in the magnitude of the climate feedback parameter at longer timescales. Conversely, simulations with higher forcing exhibit a larger feedback parameter (in magnitude). Subsequently, the evolution of surface warming patterns over multi-centennial timescales is studied. While the response is generally consistent across simulations, some variations, particularly in the South Pacific and at high latitudes, suggest a certain level of state- or forcing-dependence. The patterns of precipitation change also evolve during the stabilization runs: the drying trends found in the sub-tropical oceans and in Mediterranean-like hotspots in the SSP5-8.5 scenario tend to reduce, or even to reverse. We finally focus on the rate of heat storage in the global ocean, which is the main driver of the climate response at multi-centennial timescales. We find that the rate of warming of the deep ocean is almost independent from the amplitude of the forcing, so that most of the additional heat remains in the upper layers at high forcing. This might be due - at least partly - to a decreased ventilation of the deep ocean, caused by changes in the meridional overturning circulation (MOC). These results highlight the importance of studying multi-centennial timescales of climate change to better understand the response of the deep ocean, which will play a crucial role in determining the final state of the climate system once GHGs concentrations are stabilized.

# 1 Introduction

A key area of research in the climate community is the projection of the future state of the climate under various scenarios of anthropogenic greenhouse gases (GHG) and aerosol emissions. This global effort is coordinated by the Coupled Model Intercomparison Project (CMIP) now at its sixth phase (Eyring et al., 2016), which describes a set of common protocols followed by tens of climate modelling groups around the world. Of key importance in the CMIP6 framework are the historical simulations, forced with the observed concentrations of GHGs and anthropogenic/natural aerosols, and the scenario simulations (ScenarioMIP), which project the future state of the climate during the XXI century based on a set of prescribed GHG and aerosol concentration scenarios (O'Neill et al., 2016; Gidden et al., 2019).

The scenario simulations allow us to assess the response of the climate system to different emission pathways for the next century and, as such, are a fundamental tool for informing climate policies and impact studies. However, one limitation is that these are all transient simulations, in which the external forcing varies from year to year and the climate state is rapidly changing to adapt to the additional energy absorbed by the system. In this regard, they miss the fundamental question of what the equilibrium state of the climate system will be, once all the warming associated with a specific level of forcing has been realized.

Since the complete stabilization of the climate system extends over centuries - or even millennia, when accounting for slower components (e.g. Li et al., 2013) -, achieving this with state-of-the-art climate models poses a challenge due to their computational cost. However, exploring the *final result* of the anthropogenic perturbation to the climate system is fundamental to assess the potential for irreversible changes associated with ice sheets/glaciers melting, sea-level rise and other tipping elements (Lenton et al., 2019; Armstrong McKay et al., 2022). Indeed, when it comes to climate policies, focusing solely on impacts during the 21st century can pose a significant risk in terms of downplaying the severe climate and ecological transitions that may occur in the subsequent centuries.

In the past, mainly due to the computational constraint, multi-centennial timescales had been mostly explored with EBMs (Energy Balance Models) or EMICs (Earth system Models of Intermediate Complexity) (e.g. Zickfeld et al., 2013; Rugenstein et al., 2016a; Pfister and Stocker, 2017). In the last decade, however, a considerable number of very long integrations have also been performed with complex coupled GCMs (General Circulation Models). One remarkable example is the work by Li et al. (2013), who performed a 5000 years experiment with abrupt $4xCO_2$ forcing to achieve a complete stabilization in the deep ocean and analyse the equilibrium response of the ECHAM5/MPIOM model. Recently, the LongRunMIP protocol (Rugenstein et al., 2019) collected a multi-model ensemble of 1000-yr long climate simulations, which constitutes the most comprehensive dataset of its kind to date. The ensemble has improved the understanding of the climate response on multi-centennial timescales in many respects, including an improved estimate of equilibrium climate sensitivity (Rugenstein et al., 2020), the study of the evolution of climate feedbacks and warming patterns at long timescales (Bloch-Johnson et al., 2021; Rugenstein et al., 2020) and the ENSO response in a stabilized climate (Callahan et al., 2021).

In this work, we are mainly interested in three key aspects concerning the response of the climate system on a multi-century timescale and its dependence on the level of warming/forcing:

– the variation of the climate feedback parameter;

       – the evolution of the temperature and precipitation patterns;

       – the response of the deep ocean.

The quantification of climate feedbacks and equilibrium climate sensitivity is a very active field of research, also stimulated by the quite large spread in the available estimates both from observations and climate models (Forster et al., 2021; Zelinka et al., 2020). An additional issue is represented by the fact that climate feedbacks are not constant, as assumed in the usual linear framework (Gregory, 2004; Andrews et al., 2012), but depend on the level of global warming, on the warming pattern and on the external forcing (Andrews et al., 2015; Knutti and Rugenstein, 2015; Rugenstein et al., 2016b; Dunne et al., 2020; Mitevski et al., 2021; Bloch-Johnson et al., 2021). The evolution of the response on multi-centennial timescales has been recently explored by Rugenstein et al. (2020) and Dunne et al. (2020) in the LongRunMIP ensemble, who report a revised estimate for the equilibrium climate sensitivity when considering millennial simulations, compared to 150-yr runs. More generally, the evolution of the climate feedback may depend on the state of slower components of the climate system, like deep ocean circulation (He et al., 2017; Kajtar et al., 2021), or the efficiency of deep ocean heat uptake (Armour, 2017). In this regard, there are indications that inter-model differences in the rate of heat transfer to the deep ocean may be a key factor in driving the spread in the surface response between models (Boé et al., 2009; Kostov et al., 2014; Gjermundsen et al., 2021; Bellomo et al., 2021).

The long-term evolution of climate feedbacks is primarily influenced by changes in the global mean temperature and, if we set aside other factors, by changes in the warming pattern over time. Indeed, the warming pattern observed in the initial years following an abrupt transition differs from the one observed during the first century and a half (Andrews et al., 2015) and continues to evolve over multi-centennial timescales (Rugenstein et al., 2020). The key features include an initial strengthening (over 100 years) and subsequent weakening (over 1000 years) of the ENSO-like pattern in the Pacific, an acceleration of warming in the southern hemisphere (particularly in the Southern Ocean), and the recovery of the North Atlantic warming hole at multi-centennial timescales.

An unresolved question pertains to how these patterns depend on different levels of forcing and global warming, which might exhibit non-linear behavior (Rugenstein et al., 2016a; Mitevski et al., 2021). Regarding this specific point, Rugenstein et al. (2020) highlighted this issue but did not explore it further, due to differences in the setup across the LongRunMIP ensemble. When considering changes in the global hydrological cycle, these patterns are also expected to evolve at multi-centennial timescales following stabilization. This evolution is mainly associated with the change in the surface warming patterns and to the decreasing radiative imbalance (Wu et al., 2010; He et al., 2017; Zappa et al., 2020). However, the extent to which this response depends on different levels of warming/forcing remains unexplored.

The question of how these patterns evolve over the long term and what causes this change remains open, and carries implications for equilibrium climate sensitivity. An essential element in understanding this point lies in the pattern of ocean heat uptake and in the response of the intermediate and deep ocean layers (Armour et al., 2016). Additionally, ocean circulation plays a role in determining the amount of heat effectively transferred to the bulk ocean. However, the current understanding of

the deep ocean response to long-term warming remains limited. The topic has been explored using both EMICs (Rugenstein et al., 2016a) and complex GCMs (Li et al., 2013). Nevertheless, several issues persist, often tied to the specific formulation of physical processes in ocean models (Gregory, 2000; Exarchou et al., 2015).

We here present a set of millennial climate simulations performed with EC-Earth3, a state-of-the-art GCM participating to CMIP6 (Döscher et al., 2022). The simulations are compatible with the LongRunMIP protocol, featuring a constant external forcing for the whole integration length. A difference in the set-up with respect to most LongRunMIP contributions is represented by the fact that, instead of an abrupt increase in $CO_2$ from the pre-industrial (PI) climate, each simulation features here an *abrupt stabilization* of the external forcing starting from scenario simulations. Specifically, each simulation is branched at a specific year from the CMIP6 historical or the SSP5-8.5 simulation and the GHGs and aerosol concentrations are kept fixed thereafter at the branching-year level. The final states represent the world we would have in the far future if the atmospheric concentration of GHGs and aerosols were suddenly stabilized. Of course this is different from a sudden cease in emissions, since here we do not consider chemistry and carbon-cycle feedbacks (see for example the ZECMIP protocol, Jones et al., 2019).

Besides presenting the simulations - which are reported here for the first time -, our study aims at exploring the long-term pathway towards stabilization of the climate system in the EC-Earth3 model, with a focus on the following research questions:

1. how does the climate feedback evolve in time at multi-centennial timescales, and is it influenced by the level of warming/forcing?

2. how do temperature and precipitation patterns change in the long term, and is this evolution contingent upon external forcing?

3. how does the deep ocean respond at multi-centennial timescales? How do different forcing levels impact the deep ocean heat uptake?

The paper is structured as follows: Section 2.1 presents the model and the simulation set-up; Section 3.1 gives a general overview of some standard indices; Section 3.2 analyses the behaviour of the climate feedback parameter; Section 3.3 focuses on the patterns of warming and precipitation change at long timescales; Section 4 analyses the response of the deep ocean.

## 2 Data and methods

### 2.1 Model and simulation set-up

The simulations have been performed with the EC-Earth3 climate model (version 3.3.3), a state-of-the- art Earth-system model, which is developed by a consortium of European research institutions (Döscher et al., 2022). EC-Earth3 includes robust and validated components for the atmosphere (the ECMWF IFS model cy36r4), the ocean (NEMO 3.6; Madec, 2008), the sea ice (LIM3; Fichefet and Maqueda, 1997) and land processes (H-Tessel; Balsamo et al., 2009). It is worth to note that v3.3.3 is the same version used for the CMIP6 inter-comparison project. The model has been used in the standard CMIP6 resolution

TL255L91-ORCA1L75, corresponding to a horizontal resolution of approximately 80 km and 100 km in the atmosphere and the ocean, respectively. In the vertical, the atmosphere uses 91 levels (up to 1 hPa) and the ocean 75 levels.

Each stabilization run lasts 1000 years and is branched from the corresponding CMIP6 simulation (historical/SSP5-8.5) of EC-Earth3 (r4i1p1f1 member), following an abrupt stabilization of the GHG concentrations and of the aerosol forcing. We run a set of six simulations, corresponding to 1990 (historical), 2025, 2050, 2065, 2080 and 2100 (SSP5-8.5) conditions. In the
following, we refer to a specific simulation as "b###" with the hashes referring to the last three digits of the year of branching (i.e. "b100" for the 2100 simulation). GHG concentrations are those provided as standard input for CMIP6 (see https://esgf-node.llnl.gov/projects/input4mips/). Aerosol forcing is prescribed as optical depth following the MACv2-SP interpretation of the historical period (Stevens et al., 2017) and future CMIP6 scenarios (Fiedler et al., 2019). In terms of $CO_2$ concentrations, the six simulations correspond roughly to 1.25, 1.5, 2, 2.5, 3 and 4 times the pre-industrial value. The exact concentrations of
$CO_2$, $CH_4$ and $N_2O$ are shown in Table 1. Aerosol forcing also differs between the simulations, since the SSP5-8.5 scenario assumes that aerosol emissions will evolve during the XXI century: the main trends being a steady decrease over East Asia and an increase followed by a decrease over India and Africa, peaking in 2040 and 2080 respectively (see bottom right panel of Fig. 3 in Fiedler et al., 2019).

**Table 1.** Values of GHGs for all simulations.

|  | PI | b990 | b025 | b050 | b065 | b080 | b100 |
|---|---|---|---|---|---|---|---|
| $CO_2$ (x PI) | 1 | 1.25 | 1.5 | 2 | 2.5 | 3 | 4 |
| $CO_2$ (ppm) | 284 | 354 | 432 | 563 | 691 | 864 | 1135 |
| $CH_4$ (ppb) | 808 | 1717 | 1954 | 2446 | 2651 | 2652 | 2415 |
| $N_2O$ (ppb) | 273 | 309 | 336 | 358 | 370 | 380 | 392 |

As most current generation GCMs, this model version does not include a proper treatment of Greenland and Antarctic ice
sheets. In fact, these are represented as mountains (with fixed orography), which are covered by a 10 meter water-equivalent amount of snow. The snowfall exceeding this threshold is directly distributed as runoff to the ocean around Greenland/Antarctica, to ensure water conservation. This works quite well in a pre-industrial or present-day climate or in short scenario simulations. In our simulations, however, this simplistic representation results in a complete melting of the snow cover over Greenland and parts of Antarctica at high forcing after a few centuries, which are then left as mountains covered by bare soil. This is of
course unrealistic and drives a strong local surface warming in Greenland and Antarctica in the high forcing simulations (in particular in b100), which should then be interpreted accordingly.

## 2.2 Definitions and procedures

In this Section, we provide definitions and detail the methods used throughout the paper. The variables considered are part of standard CMIP6 output for the atmospheric (2-meter air temperature, precipitation, shortwave (SW) and longwave (LW)

radiation at top-of-atmosphere (TOA) and at the surface, surface sensible and latent heat fluxes, snowfall and snow melt) and ocean component (sea-ice concentration, 3D conservative temperature, overturning mass streamfunction).

**Stabilization timescale**

The duration of the surface temperature stabilization is estimated in Section 3.1 (Table 2) through a series of Mann-Kendall (MK) trend tests, with the following iterative procedure:

1. an MK test is performed from $y_0$ to the end of the simulation, starting from $y_0 = 0$;

   2. if the test indicates a significant increasing trend, $y_0$ is moved forward by 10 years and step 1 repeated;

   3. if the test indicates an insignificant trend or a significant decreasing trend, $y_0$ is taken as the estimate of the stabilization period.

If condition 3 is not satisfied and $y_0 > 950$ years, then no estimation of the stabilization is given, since the remaining time is too
short to assess a trend (this is the case of b050 in Table 2). Considering the estimates in Table 2 for the surface stabilization, we refer throughout the paper to timescales of the order of the stabilization period (multi-centennial) as "long" or "stabilization" timescales, and at shorter timescales, up to one century, as "short" or "transient" timescales.

**Net radiative fluxes and spurious energy imbalances**

The net radiative flux at TOA is computed as $N_{TOA} = SW_\downarrow - SW_\uparrow - LW_\uparrow$. Since the EC-Earth3 model presents a small
spurious energy imbalance, the net TOA flux does not converge at zero at full equilibration. The spurious energy imbalance of the atmospheric component of EC-Earth3 has been estimated as 0.25 W/m$^2$ (energy source) in (Döscher et al., 2022), analyzing a pre-industrial control run. Since the cause is currently unknown, this imbalance is not assured to be independent from the climate state (warming/forcing), as observed for many models (Sanderson and Rugenstein, 2022). We check for this by comparing the net TOA flux with the net surface flux in all simulations. Since the atmospheric heat capacity is about three
orders of magnitude less than the ocean heat capacity, the net TOA and net surface fluxes should equilibrate in the global and annual mean. Their difference is indicative of a spurious sink/source of energy inside the atmospheric component. The surface net energy flux is computed as:

$$N_{\text{srf}} = SW_\downarrow - SW_\uparrow + LW_\downarrow - LW_\uparrow - H_\uparrow - L_\uparrow - \left( S_{fall}|_{oce} + S_{melt}|_{land} \right) \lambda_{melt}$$

Where $H_\uparrow$ and $L_\uparrow$ are the sensible and latent heat fluxes at the surface, $S_{fall}$ is the snowfall flux, $S_{melt}$ is the amount of snow
that gets melted (and goes to runoff), $\lambda_{melt}$ is the specific latent heat of melting. The snowfall contribution needs to be taken into account since it constitutes a negative flux of latent energy from the atmosphere to the surface, which is not accounted for in the latent heat flux. Over land, the IFS model already considers the snow sublimation energy inside the latent heat flux $L_\uparrow$, so only the fraction that goes to runoff ($S_{melt}|_{land}$) needs to be taken into account.

We define the intrinsic atmospheric imbalance as $I_{atm} = N_{\text{srf}} - N_{TOA}$. For the ocean, the imbalance is computed comparing the oceanic surface heat flux (as seen by the atmosphere) to the effective ocean heat uptake (derived from the variation of the total energy content of the ocean, see paragraph Ocean heat content below): $I_{oce} = \frac{dE_{oce}}{dt} - N_{\text{srf, oce}} - F_{GH}$. Also the bottom geothermal heat flux $F_{GH}$ - which is constant in our simulations - is taken into account.

We report in Section 3.2 on the intrinsic atmospheric and oceanic imbalances. Then, we correct the net TOA flux and the net surface flux over the ocean for the respective imbalances, such that:

$$N'_{TOA} = N_{TOA} + I_{atm} + f_{oce} \cdot I_{oce}$$

$$N'_{\text{srf, oce}} = N_{\text{srf, oce}} + I_{oce}$$

Where $f_{oce}$ is the fraction of the global surface covered by oceans.

**Climate feedback parameter**

Following the common linear framework (Gregory, 2004; Andrews et al., 2012), the response of the climate system to an external radiative forcing $F$ can be seen, at first instance, as:

$$N = F + \lambda \Delta T_s$$

Where $N$ is the net radiative imbalance at TOA, $F$ is the forcing imposed, $\Delta T_s$ is the change in the system temperature and $\lambda$ is the climate feedback parameter.

The climate feedback parameter is estimated in the following as the slope of the regression of global net incoming TOA radiation with respect to GTAS (Gregory, 2004; Andrews et al., 2012). Both variables are averaged in 10-year windows before the regression, to avoid confounding the long-term climate feedback with faster feedbacks linked with the interannual variability (see e.g. Proistosescu et al., 2018). This is particularly important when dealing with a small residual imbalance, which would otherwise be dominated by the interannual variability (this is the case for b990 and b025 for example, and of all simulations close to the end of the run). Using a shorter or longer window (e.g. 5, 20 years) gives very similar results. Net incoming TOA has been corrected for imbalances before the computation (see above Section).

**Temperature/Precipitation patterns**

The temperature patterns discussed in Section 3.3 (Figure 7) are calculated as a linear regression of the local surface temperature against the 50-year lowpass-filtered global mean 2-meter air temperature (GTAS), using the whole time range of the simulations. The GTAS is lowpass-filtered to highlight the long-term patterns of change rather than the interannual and decadal variability patterns (in particular the ENSO pattern), which can be of the same or higher order of magnitude for low warming scenarios. Precipitation patterns in (Figure 8) are calculated in an analogous way, apart from the fact that the local precipitation field is first divided by the local pre-industrial mean field to obtain the relative precipitation. The precipitation pattern is masked where the pre-industrial average total annual precipitation is less than 50 mm/year, to avoid over-emphasizing changes in desert areas. The significance of the changes is assessed through a Wald test.

We stress here that, for the stabilization simulations, the trends are calculated starting from the stabilization year, and thus don't include changes happened before branching, during the transient SSP5-8.5 simulation. This is done to highlight how the patterns evolve during the stabilization with respect to the transient SSP5-8.5 simulation. Therefore, the interpretation should be careful, for example, regarding the inversion of the precipitation trends (Figure 8): in most cases, the stabilization trend can locally reduce the transient response, but the amplitude is generally not enough to reverse the sign of the total change with respect to pre-industrial.

**Ocean Heat Content**

We calculated the additional heat stored in an ocean layer between $z_1$ and $z_2$ as:

$$\Delta H_{12} = \sum_{z_1}^{z_2} c_p \int_{A_z} m_A \Delta\Theta_c \, dA$$

where $\Delta\Theta_c$ is the change in conservative temperature (McDougall, 2003) with respect to pre-industrial, $c_p$ is the specific heat capacity, $A_z$ is the area covered by level z and $m_A$ the mass per unit area of that level. The conservative temperature is usually very similar to the potential temperature, and indistinguishable for most purposes. However, the former is more indicated for heat budget analysis (McDougall, 2003). Since the simulations are branched from the r4i1p1f1 member of the CMIP6 SSP5-8.5 simulation, which derives from the r4 member of the CMIP6 historical, the pre-industrial state is taken as the 50 years that anticipate the branching of the r4 historical simulation.

**Meridional Overturning Circulation indices**

The streamfunctions of the Meridional Overturning Circulation for the global, Atlantic and Pacific/Indian oceans are available as model outputs. We define here the Atlantic MOC (AMOC) index as the maximum of the Atlantic MOC streamfunction between 30 and 50 N and 500 and 2000 m depth. The abyssal Southern MOC (SMOC) index is defined here as the average value of the global MOC streamfunction between 30 and 50 S and 3000 to 4000 m depth. The average is taken rather than the maximum to reduce the fluctuations in the index.

## 3 Results

### 3.1 Overview

We give here a general overview of the simulations, analysing the global mean response to different levels of stabilized forcing. Figure 1 shows the GTAS anomaly with respect to the pre-industrial climate for the six runs. The warming continues in all experiments well after the abrupt stabilization of the GHG concentrations, and the final anomaly ranges between 1.4 to 9.6 K above the pre-industrial mean climate. Values for each experiment are indicated in Table 2. Even the relatively low forcing b025 simulation well exceeds in the long term the Paris agreement goals of 1.5 and 2 degrees warming above pre-industrial, while b990 would represent the only stabilized climate consistent with those goals. As a first estimate, this shows that, for the

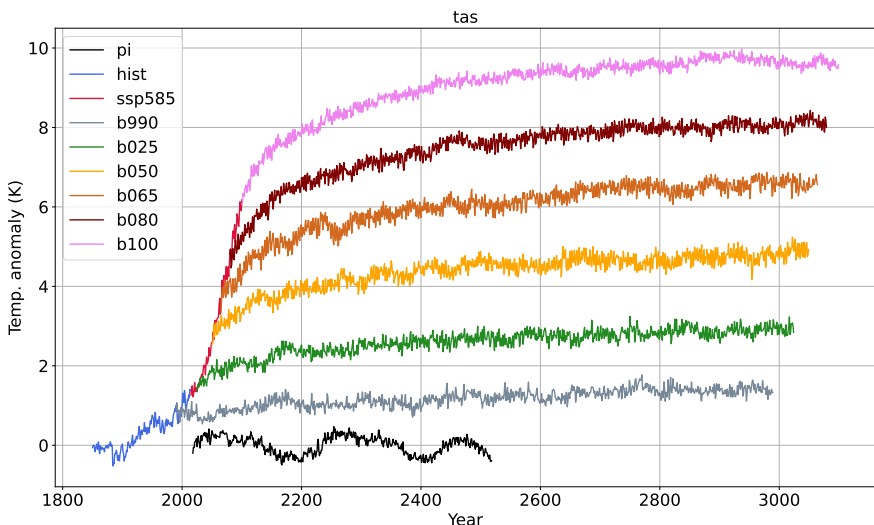

**Figure 1.** Global mean surface air temperature (GTAS) anomaly of the historical (hist), SSP5-8.5, b990, b025, b050, b065, b080 and b100 with respect to the pre-industrial mean climate.

EC-Earth model, the target $CO_2$ concentration in the atmosphere needed to land in a desirable stabilized climate is much lower

than the scenario concentration for 2025 (432 ppm), and very close to the 1990 concentration (354 ppm). The EC-Earth3 model has an effective climate sensitivity of 4.10 K, which is very close to the CMIP6 multi-model average of 3.9 K (Zelinka et al., 2020). Apparent from Figure 1, it is the centennial oscillation in the pre-industrial temperature (black line), which is driven by internal variability of the Atlantic Meridional Overturning Circulation (AMOC) (Meccia et al., 2022). Also the stabilization simulations feature internal variability in GTAS at multi-decadal timescales, but with relatively smaller amplitude and period

(see Meccia et al. (2022) for a discussion of the AMOC variability in b050).

An estimate of the timescale for the stabilization of the surface climate is given in Table 2 by applying the Mann-Kendall test to the GTAS timeseries (details on the method can be found in Section 2.2). The global mean surface temperature stabilizes in most cases before the end of the simulation, with the notable exception of b050, which still has a significant positive trend at the end. Another way to look at the residual changes at the surface is through the residual increase in GTAS in the last 500

years, which is of the order of a few tenths of a degree for all simulations. Interestingly, despite the extreme forcing, the b100 simulation has the shortest surface stabilization period (even shorter than b990), and features a residual temperature increase in the last 500 years which is less than that in the intermediate b050, b065 and b080 simulations. Apart from b050, all simulations reach at the end what Li et al. (2013) call "quasi-equilibrium", or a state in which the change in GTAS is very small, while the deep ocean heat uptake goes on. In this condition, the surface layer is equilibrated (at least on a global scale), and the residual

net TOA is completely absorbed by the deep layers of the oceans, which are still out of equilibrium.

**Table 2.** Total ΔGTAS at the end of the simulation with respect to pre-industrial, ΔGTAS of last 500 years, final corrected net TOA flux (considering 30-year periods) and estimated surface stabilization period (see Section 2.2).

|  | b990 | b025 | b050 | b065 | b080 | b100 |
|---|---|---|---|---|---|---|
| ΔGTAS (K) (final) | 1.4 | 2.9 | 4.9 | 6.6 | 8.1 | 9.6 |
| ΔGTAS (K) (last 500) | 0.1 | 0.2 | 0.4 | 0.5 | 0.5 | 0.3 |
| net TOA (W/m$^2$, corr.) | 0.14 | 0.23 | 0.22 | 0.33 | 0.25 | 0.33 |
| surface stabilization (years) | 690 | 780 | - | 800 | 910 | 650 |

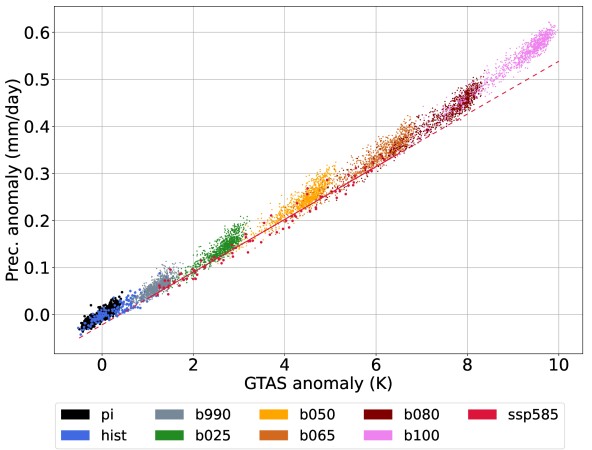
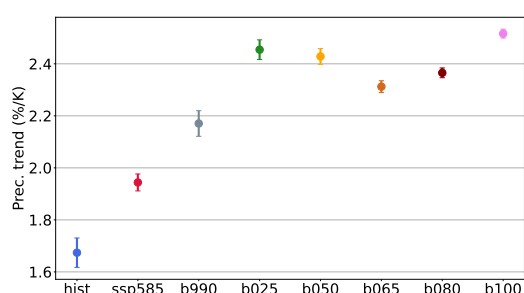

**Figure 2.** Left panel: Global annual mean precipitation anomaly with respect to the GTAS anomaly with respect to pre-industrial. The red line is a linear fit to the SSP5-8.5 data. Right panel: trend of the global annual-mean relative precipitation change with respect to GTAS for each simulation.

Figure 2 shows the annual global mean precipitation against GTAS for all runs. The trend is close to linear during the transient SSP5-8.5 simulation, as shown by the linear fit to the red dots in Figure 2 (left panel). An apparent feature of the stabilization simulations is an intensification of the hydrological cycle with respect to the transient response, implying more precipitation for the same global warming level. This is likely due to a larger weakening of the hydrological cycle in the
SSP5-8.5 scenario due the rapid adjustments to GHG forcing, whose response scales with GHG concentrations rather than with surface mean temperature (Samset et al., 2016). Indeed, the transient SSP5-8.5 simulation shows a precipitation increase of about 1.9% per 1 K of global warming, slightly larger than the 1.7%/K trend during the historical period (right panel of Figure 2). The stabilization runs all show larger trends, from 2.2 to 2.5% per degree of warming.

Figure 3 shows the winter and summer sea-ice extent for the Arctic (upper panels) and for the Antarctic region (bottom).
Arctic winter sea-ice extent decreases progressively in all simulations, stabilizing towards the end of the period for most runs,

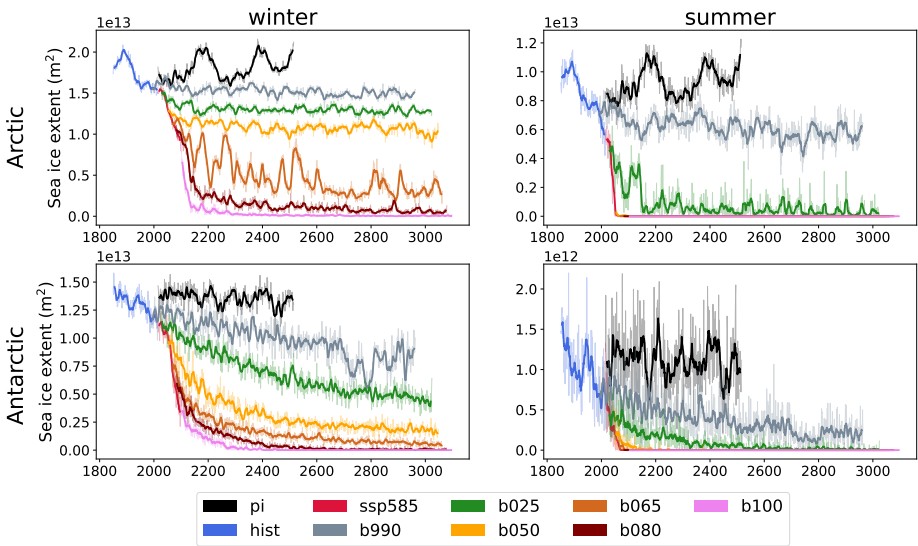

**Figure 3.** Upper panels: Arctic sea-ice extent at the end of the local winter (March) and summer (September) seasons. Bottom panels: Antarctic sea-ice extent at the end of the local winter (September) and summer (March) seasons. The thin lines show the monthly mean values, the thick ones are 10-year running means.

with the notable exception of b065. b100 simulates an abrupt collapse just a few years after the branch-off from SSP5-8.5: this behaviour has already been observed in an extended RCP8.5 scenario in a previous version of EC-Earth (Meccia et al., 2020) and in other climate models (Bathiany et al., 2016; Drijfhout et al., 2015). b065 has apparently reached a tipping point in the winter Arctic sea-ice and features a strong instability in the whole simulation, switching periodically from a low to a high ice 265 cover state. In summer, the Arctic reaches a completely ice-free state in all simulations shortly after the branch-off, except for b990 and b025. The latter also shows an almost complete collapse, although with significant interannual variability. Only b990 retains a considerable fraction of Arctic summer sea-ice. Antarctic sea-ice experiences a gradual summer decline in b990 and b025, while all other simulations reach a completely ice-free state during the transient or at the beginning of the stabilization period. As for the summer Arctic sea-ice, eventually b025 also shows an almost complete collapse of summer Antarctic sea-ice 270 cover. The winter Antarctic sea-ice cover experiences a gradual decline during the course of all simulations, reflecting the delayed warming of the Southern ocean. b080 and b100 eventually reach an ice-free state, b050 and b065 stabilize in the second half of the simulation retaining a small fraction of the original sea-ice cover, while b990 and b025 are still drifting towards their new equilibrium at the end of the simulation.

### 3.2 Forcing dependence of the climate feedback parameter

The path to equilibrium of the simulations can be appreciated in Figure 4 (left panel), which shows a scatterplot of the global net energy imbalance at TOA ($W/m^2$) with respect to GTAS, averaged every 10 years. The imbalance ranges initially from about

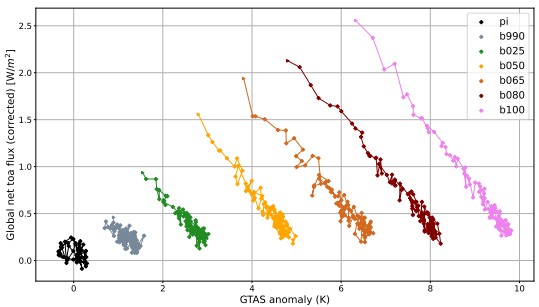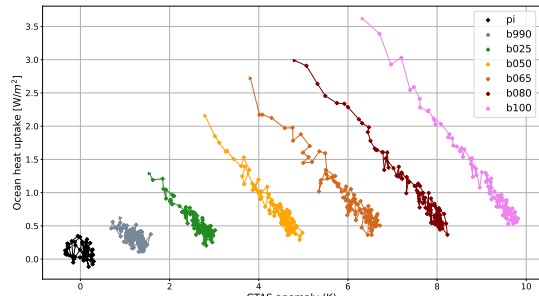

**Figure 4.** Left panel: global net energy imbalance at TOA (corrected for spurious model energy sources) vs GTAS anomaly of the different simulations. Right panel: global surface ocean heat uptake vs GTAS anomaly. Dots represent 10-year averages.

0.4 W/m$^2$ for b990 to about 2.5 W/m$^2$ for b100, and then slowly approaches zero during the course of the simulation. The right panel of Figure 4 shows the global ocean heat uptake against GTAS. The global net TOA flux has been corrected to account for the spurious internal energy production in the atmospheric and oceanic components (see Section 2.2 for details). The IFS

model features a small spurious energy source, which was estimated to be about 0.25 W/m$^2$ in the pre-industrial control run (Döscher et al., 2022). We report here that this spurious energy source in IFS is partially dependent on forcing/warming, with our estimates ranging from about 0.3 W/m$^2$ for the pre-industrial control to 0.75 W/m$^2$ at the end of the b100 simulation (see Table 3). This fact should be further explored and taken into account when studying the system response at high forcing, as recommended by Sanderson and Rugenstein (2022). The extent to which this could impact the estimates of climate sensitivity

is expected to be proportional to the ratio $\Delta I_{atm}/F$, which is about 6% for b100 ($F_{b100} \approx F_{4\times CO_2}$).

**Table 3.** Spurious energy production in the atmospheric (IFS) and oceanic (NEMO) components (global mean, W/m$^2$), average for the last 100 years of the simulations.

| Comp. | pi | b990 | b025 | b050 | b065 | b080 | b100 |
|---|---|---|---|---|---|---|---|
| IFS | 0.30 | 0.34 | 0.39 | 0.46 | 0.51 | 0.59 | 0.75 |
| NEMO | -0.48 | -0.48 | -0.49 | -0.49 | -0.48 | -0.47 | -0.48 |

The spurious energy sink in the NEMO model is instead constant across different forcing levels, about 0.5 W/m$^2$.

We further observe that the slope of the equilibration pathways tends to be more negative for high levels of forcing/warming. This might be due to the fact that the feedback parameter $\lambda$ is not a constant, but depends on the global warming level, on the warming pattern and on the external forcing imposed (Bloch-Johnson et al., 2021). At intermediate levels of warming, a likely

contribution to this non-linear behaviour is the sea-ice albedo feedback, which decreases for increasing warming, reaching zero for b100. Possibly other feedbacks contribute as well, including cloud feedbacks, which may depend on the warming level and on the pattern of warming (Rugenstein et al., 2016b). It is out of the scope of this paper to undertake an in-depth analysis of the

individual climate feedbacks, but we provide here some more insight on the variation of the net climate feedback parameter across the simulations.

Figure 5 shows the climate feedback parameter estimated for the first (empty circle) and second (diamond) half of each simulation (for details on the procedure, see Section 2.2). Two interesting points are noted:

– during the stabilization, the feedback parameter tends to become less negative, due to the dependence on GTAS and on the changing warming pattern, consistently to what other works have shown on these timescales (Rugenstein et al., 2020; Bloch-Johnson et al., 2021);

– the feedback parameter tends to be more negative for larger forcing/warming. The relation is not linear, but is generally confirmed both for the first and for the second half of the simulations. This seems the opposite of what found in Bloch-Johnson et al. (2021) for most models, regarding the dependence of the feedback parameter both on temperature ($\delta_T \lambda$) and on the $CO_2$ concentration ($\delta_C \lambda$), although still consistent with one model in that study.

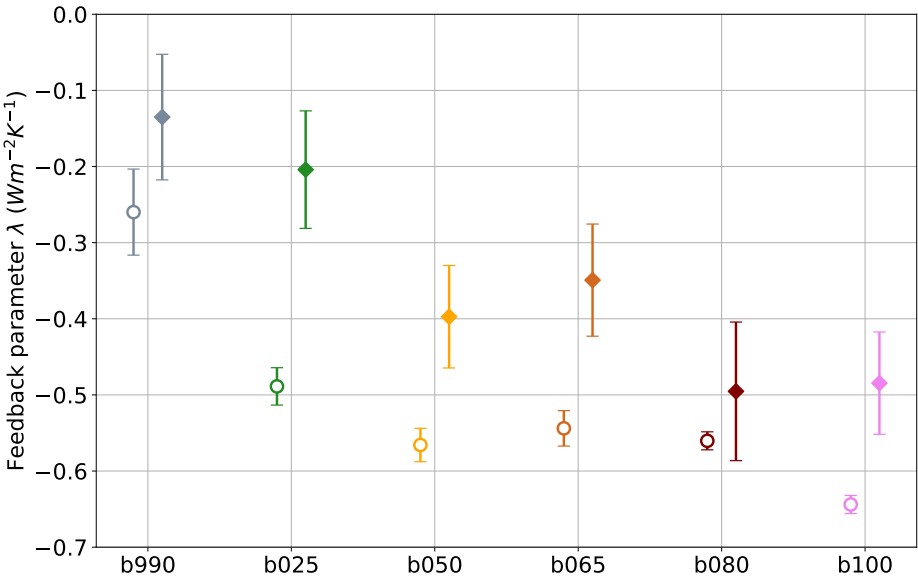

**Figure 5.** Climate feedback parameter estimated for the first (empty circle) and last (diamond) half of the simulations.

However, we are cautious against a direct comparison with Bloch-Johnson et al. (2021) due to differences in our experimental
setup, regarding both the initialization and the external forcing. In fact, it is worth noting that our stabilization simulations start from different climate states, rather than being abrupt perturbations to a pre-industrial climate as in Bloch-Johnson et al. (2021), Mitevski et al. (2021) and Rugenstein et al. (2020). However, it is unlikely that the starting point is playing a role here, since the

behaviour is generally confirmed also for the second half of the simulations. Moreover, if this was the case, we would expect the scenario that spent a longer time in the transient (b100) to show a less negative feedback, which is the opposite of what we observe. Also, in our case, the external forcing includes contributions other than $CO_2$, namely other GHGs and aerosols. While we expect that the contribution of other GHGs roughly aligns with that of $CO_2$, aerosols may exert a different impact, especially on the lower forcing cases. However, it is improbable that this factor alone can explain the observed difference in the feedback parameter. Such an explanation would imply an exceedingly high sensitivity of feedbacks to the forcing agent, whereas recent research indicates a moderate sensitivity at most (Salvi et al., 2022; Richardson et al., 2019).

We plan to further explore in a future work the apparent inconsistency between our model's result and previous results (Bloch-Johnson et al., 2021; Meraner et al., 2013), analysing the individual feedbacks contributions and the role of different forcing agents.

### 3.3 Long-term evolution of temperature and precipitation patterns

We now analyze the spatial patterns of change in the stabilization simulations. Figure 6 shows the final surface temperature anomaly (last 30-years) for each simulation with respect to the pre-industrial climate. Historical and SSP5-8.5 simulations are also shown for comparison. The analogous maps of final change for precipitation are shown in the supplementary Figure S1.

To assess in which respect the stabilization trends differ from the transient ones, we computed warming patterns (local temperature change per degree of global warming) for all simulations, as described in Section 2.2. The complete set is shown in Figure S2, and we show in Figure 7 (top row) only the SSP5-8.5, b025 and b100 patterns. The bottom row of Figure 7 shows instead the ratio of the stabilization warming patterns to the transient one, to better highlight the differences. As expected, the patterns differ in various respects between the transient and stabilization simulations. First, the SSP5-8.5 simulation features a strong asymmetry of warming between land and oceans, which is still present but much reduced in b025 and almost disappeared in b100 (apart from the northernmost regions of America and Siberia).

The longer timescales of ocean warming with respect to land warming are linked to the large thermal inertia of the global ocean, and determined by the efficiency of heat transfer at depth (Joshi et al., 2008; Boé and Terray, 2014; King et al., 2020; Lee et al., 2021). The relative warming of land and ocean regions at equilibrium is part of a more complex set of interactions where several mechanisms come into play, including lapse-rate and humidity feedbacks (Byrne and O'Gorman, 2013) and ocean circulation dynamics (Long et al., 2014).

Another difference regards the intensity and extent of the amplification at the extreme northern latitudes. If b025 still features a strong Arctic amplification, similar to the SSP5-8.5 one, this is much reduced in b100, since most of the warming there already took place during the transient. In fact, as seen in Figure 3, both the Arctic and Antarctic sea-ice cover keep reducing during b025, while b100 is already almost sea-ice-free shortly after the beginning. The extreme inland warming of Greenland in b100 is spurious, since the model does not include a proper land-ice component, as explained in Section 2.1. The North Atlantic subpolar gyre also shows an interesting behaviour, with an intensified warming during stabilization, more evident in b025 (Figure 7, bottom row), and likely linked to the recovery of the AMOC (see Figure 13).

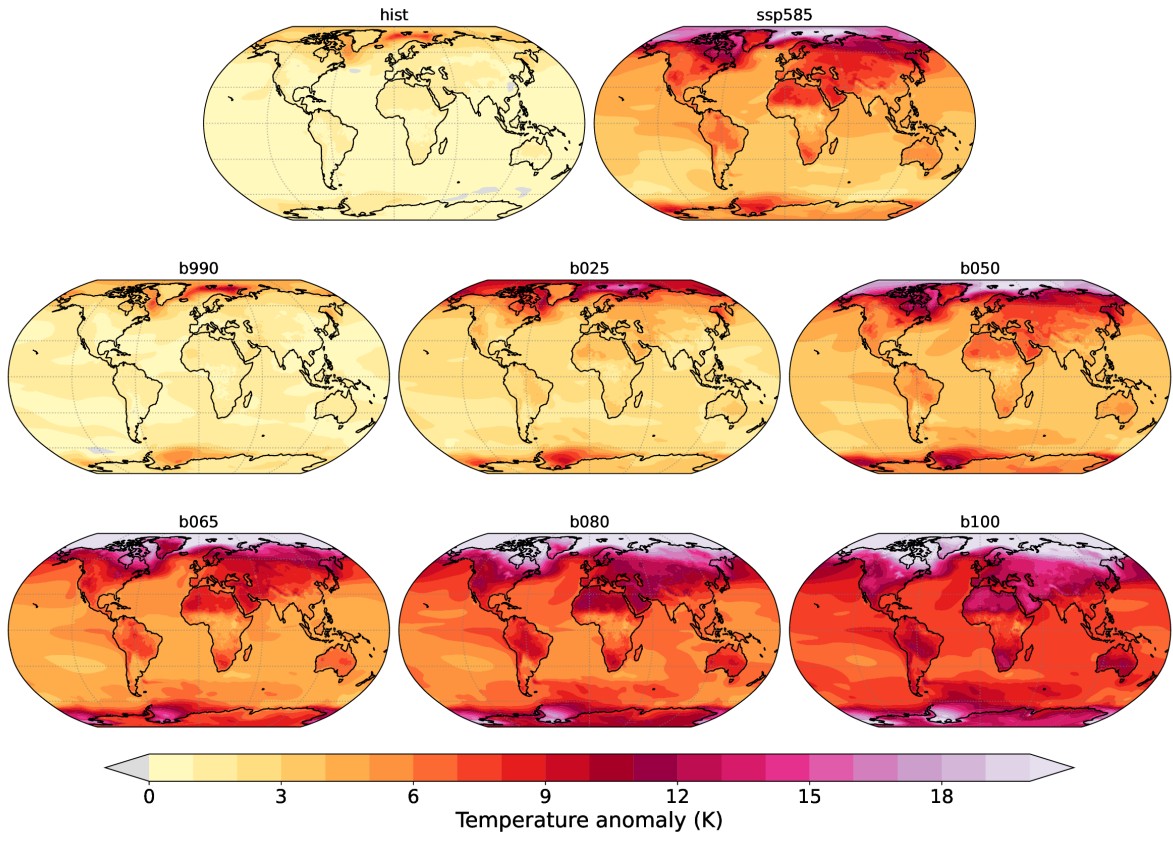

**Figure 6.** Temperature anomaly with respect to PI climate at the end of each simulation (30 yr mean).

As already reported by Li et al. (2013) and Rugenstein et al. (2020), the most apparent difference of the multi-centennial warming trend is in the Southern Hemisphere, which lags the global warming during the SSP5-8.5 simulation. The lag in the Southern ocean is quite well understood and linked to the upwelling of cold waters in the Southern Ocean (Armour et al., 2016). Also in our simulations, the warming trend in the Southern Ocean drastically increases at longer timescales, being first mostly concentrated to the South of the Atlantic and Indian oceans at low warming (b025) and then extending to the Pacific region as well (b100). The increased warming in b100 also shifts northwards, reaching inland Southern South America and Australia. As for Greenland, also the inland warming of Antarctica in b100 is unreliable due to the lack of a land-ice component. As a consequence of the increased warming in the Southern hemisphere and decreased warming in the Northern one, the global warming pattern is progressively more symmetric in the North-South direction across the stabilization runs. The region with minimum trend moves accordingly from the Southern ocean in the transient simulation to the tropical and sub-tropical oceans. These results are broadly consistent with the multi-model mean response in Rugenstein et al. (2020), in particular with regards to the Southern ocean amplification and to the increased warming in the North Atlantic.

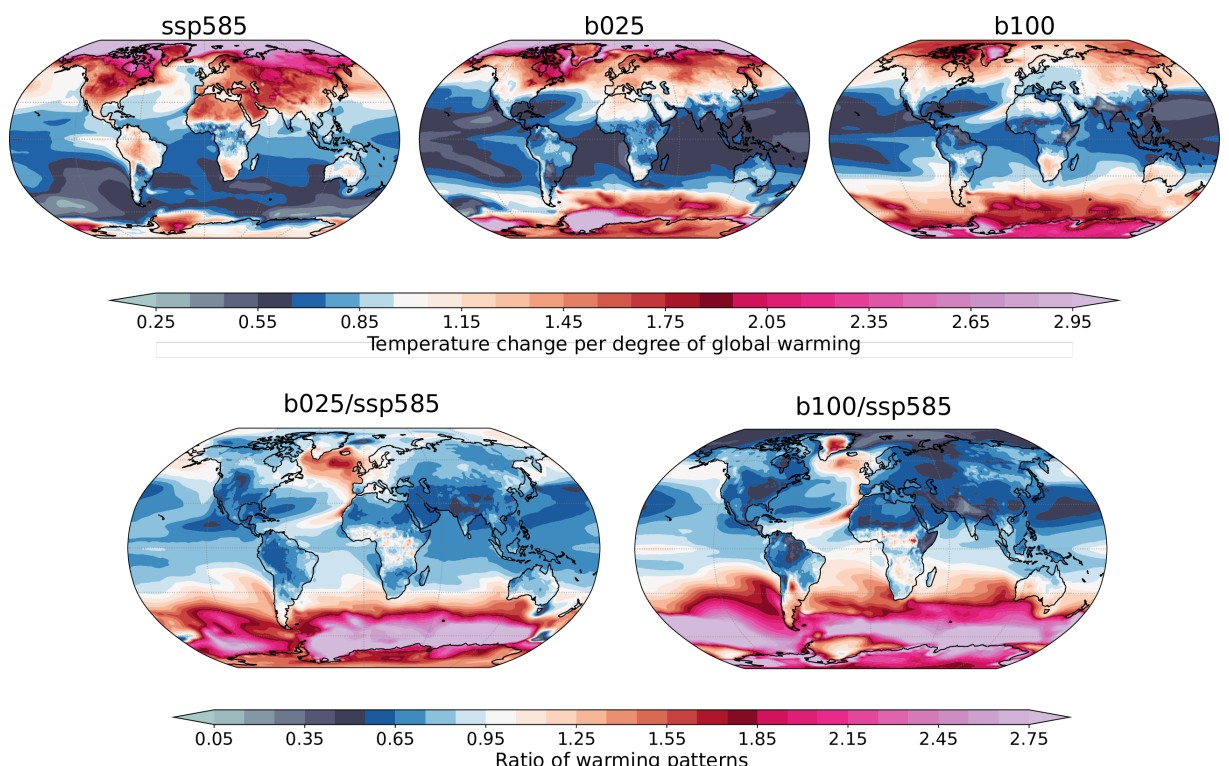

**Figure 7.** Top row: local surface temperature change per degree of global warming during the course of the simulation, for SSP5-8.5, b025 and b100. A value smaller (larger) than one means that the local rate of warming is slower (faster) than the global mean. All trends are significant at the 95% level. Bottom row: ratio of the stabilization warming patterns (b025, b100) to the transient one (SSP5-8.5).

The differences observed here between the transient and stabilization responses are consistent with the fast and slow components of the climate response as analyzed by Long et al. (2014) and Held et al. (2010). Compared to the fast response pattern, the tendencies in the slow component are reversed in regions with strong dynamical coupling to the deep ocean, in particular the North Atlantic and Southern Ocean (Long et al., 2014). We observe here how the extension of such regions and the amplitude of the response are also sensible to different levels of forcing, being more widespread for stronger forcing levels. This is relevant on a mitigation perspective, since regions that feature an intensification of the response at longer timescales are exposed to irreversible changes even in radical climate stabilization scenarios (Held et al., 2010; Kim et al., 2022).

To better assess the dependence of the warming patterns on different levels of warming/forcing, we provide a synthetic assessment in Figure S4. The Figure shows the regression of the warming patterns for each simulation with respect to the logarithm of the $CO_2$ concentration, which is roughly proportional to the effective external forcing (neglecting here other GHGs and the effect of aerosols). The intensification of the warming in the South Pacific for larger forcing is confirmed, along with a general increase in the Southern hemisphere response. Subpolar ocean regions show a decrease of the warming for increasing forcing: this is most probably due to the disappearance of sea-ice in the warmest simulations.

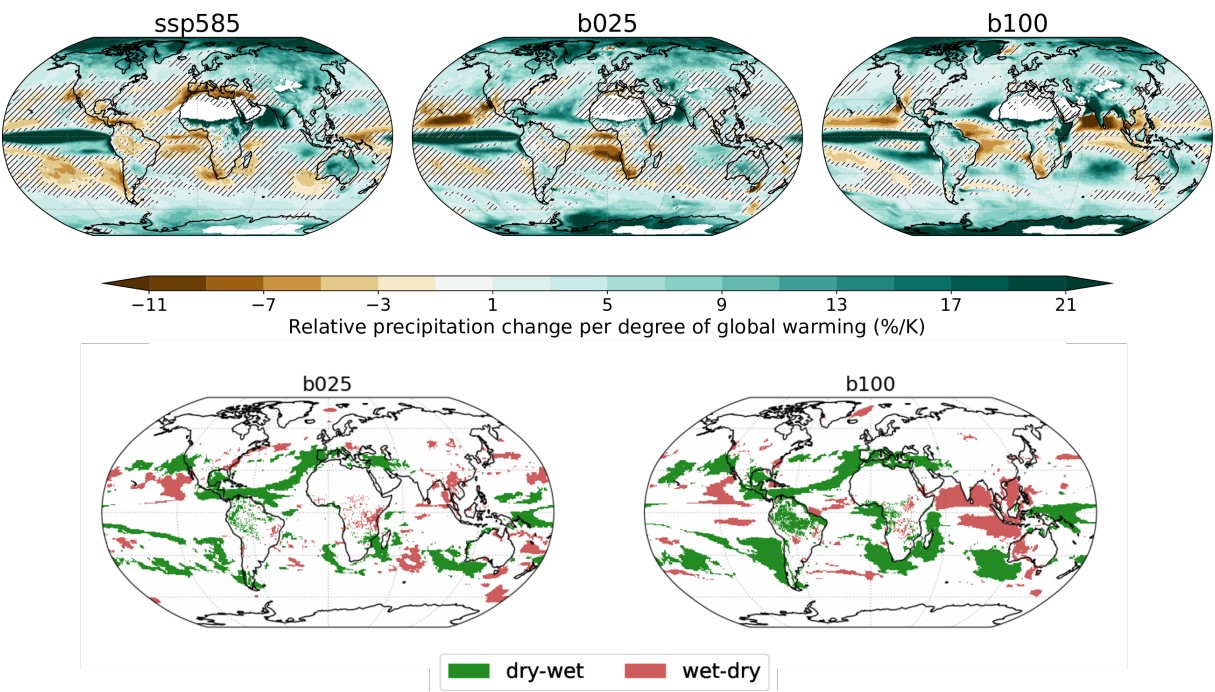

**Figure 8.** Top row: relative precipitation change per degree of global warming during the course of the simulation, for SSP5-8.5, b025 and b100. Hatching indicates regions where the trends are not significant at the 95% level. Bottom row: regions where the sign of the precipitation trend reverses during stabilization (b025/b100) with respect to the transient (SSP5-8.5) are highlighted. Dry-wet transition is shown in green, wet-dry in red. Regions with less than 50 mm/year in the pre-industrial climate are masked.

We now focus on the evolution of precipitation patterns during stabilization. Figure 8 shows the pattern of relative precipitation change per degree of global warming (top row) for SSP5-8.5, b025 and b100 (the other simulations are shown in Figure S3). Regional trends differ significantly between the transient and stabilization simulations, also changing sign in some cases. The bottom row of Figure 8 highlights regions that turn from a drying trend during SSP5-8.5 to a wettening during stabilization (green) and the opposite condition (wettening → drying, in red). For the highlighted regions, the final change with respect to the pre-industrial climate is partially mitigated and less severe than at the beginning of the simulations. However, the stabilization change is usually smaller than the transient one and the sign of the total change with respect to pre-industrial is generally consistent with that of the transient response (see Figure S1). It is worth noting that for some of the areas highlighted in the bottom row of Figure 8 the trends during stabilization are non-significant due to the small signal-to-noise ratio, especially at low forcing, so the results may be partly biased by internal variability.

Drying trends found in the sub-tropical oceans and in Mediterranean-like hotspots during transient tend to reduce during stabilization, or even to reverse. This is particularly clear for the subtropical North-Atlantic and central America, which experience drying during SSP5-8.5 and wettening in b025 and b100, and also seen in the Mediterranean region, although the

stabilization trends are not significant there. This long-term relaxation of the drying over the Mediterranean region was also observed for the ECP4.5 extended scenario of CMIP5, up to 2300 (Zappa et al., 2020).

The South-eastern Pacific and the Pacific coast of South America also experience drying in the transient and partly in b025, but wettening in b100. Overall, this is also in line with Zappa et al. (2020), who found no trend over Chile in the extended ECP4.5 scenario. However, we observe that the long-term trend in the region is also partly dependent on the level of global warming. This is also true for the Southern sub-tropical oceans, namely the ocean around South Africa and off the south-western coast of Australia, that turn more markedly to wettening in b100. The changes are however mostly over the oceans in this case (He and Soden, 2017), with the inland drying in South Africa still going on in b025 and b100. The California peninsula experiences an inversion of the transient trends, that turn to wettening in the northern part (consistent with Zappa et al. (2020)) and drying in the southern peninsula.

b100 also shows widespread changes in the tropics, including a significant inversion of the drying trend over Amazonia (that turns to wettening), increased wettening of the Horn of Africa and pronounced drying in the western tropical Atlantic, in the northern Indian ocean and in the South East Asian sea. This is likely connected with global changes in the Walker circulation system that emerge over long timescales and are more evident with high forcing. Following Chadwick et al. (2013), apart from direct radiative effects linked to GHG concentrations, the precipitation in the tropics is mainly controlled by two processes. On one side, the decrease in the tropical convective updraught drives a weakening of the Walker circulation and acts to reduce the hydrological sensitivity with respect to Clausius-Clapeyron scaling (Held and Soden, 2006). On the other side, the evolution of the SST warming pattern at longer timescales - linked to ocean circulation and warming at depth - shifts tropical convective regions, modifying the equilibrium response with respect to the transient period (Chadwick et al., 2013). In this respect, the increased warming of the southern subtropics in b100 - not seen at smaller forcing - may be playing a role in the different tropical precipitation response.

As for the temperature, we also provide a synthetic assessment of the warming/forcing dependence of the precipitation patterns in Figure S5. This analysis confirms that the precipitation increase in the Southern subtropics is intensified at larger forcing, as well as the drying in the eastern Indian ocean. The wettening trend in the Mediterranean region is instead reduced at larger forcing.

## 4   Response of the deep ocean

The long-term scales of climate stabilization are primarily driven by the ocean thermal inertia and by the efficiency of heat transfer towards the deeper layers of the ocean. Figure 9 shows the additional heat stored globally in three ocean layers: the upper layer down to 700 m depth, the mid layer from 700 to 2000 m and the deep layer below 2000 m. The contributions of the three layers are stacked one on top of the other, so that the cumulative curve shows the total heat absorbed by the global ocean during the course of the simulation. As expected, the total amount of heat stored is roughly proportional to the forcing strength, and it is still increasing at the end of the simulations in all cases. However, the rate at which the heat is stored in the three layers differs depending on the forcing. The mid and upper ocean layers show marked differences both in the amount

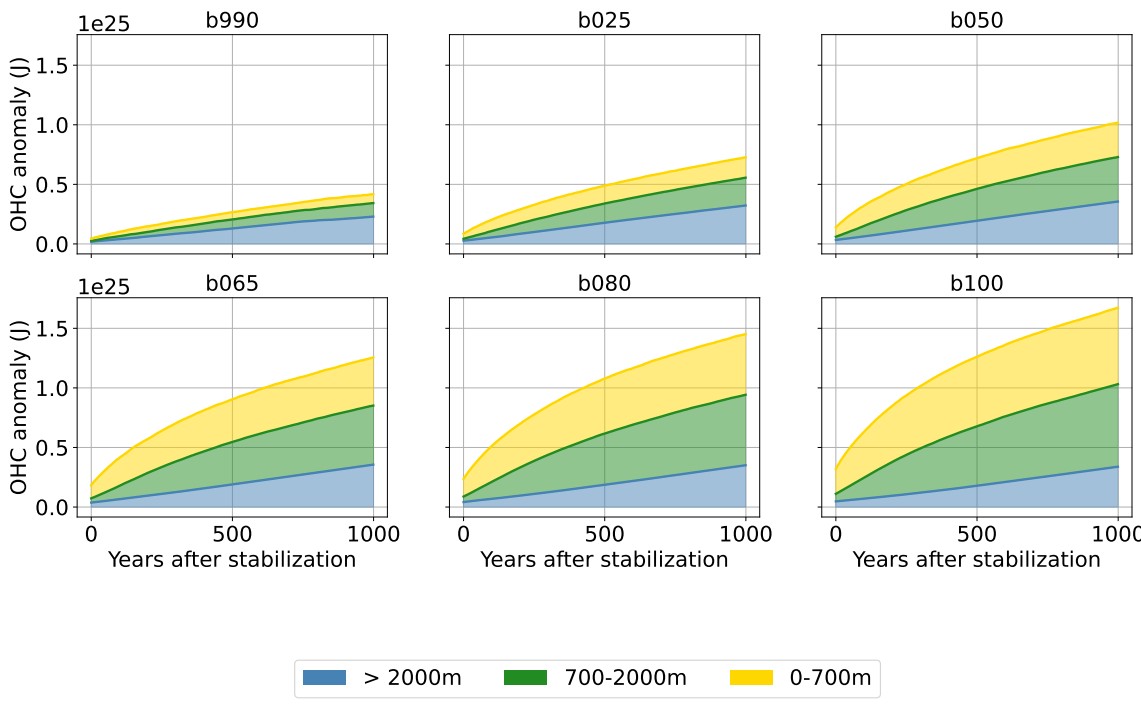

**Figure 9.** Cumulative values of global ocean heat content anomaly for all stabilization simulations in three depth regions: 0-700m, 700-2000m and below 2000m. The contributions of the three layers are stacked one on top of the other, so that the cumulative curve shows the total heat absorbed by the global ocean. The relative contributions of the three layers are shown in supplementary Figure S6.

and in the rate of heat storage. Specifically, the trend is roughly linear for the upper and intermediate ocean layers in b990 and b025, while an exponential relaxation behaviour is apparent in the simulations with larger forcing. Remarkably, the deep layer heat storage is pretty linear with time in all simulations and almost independent from the external forcing.

**Table 4.** Additional heat stored globally in upper, mid and deep ocean layers at the end of the simulations (30-yr averages, values are in units of $10^{24} J$). Relative values (percentage of heat stored in each layer) are reported in Table S1.

| Layer | b990 | b025 | b050 | b065 | b080 | b100 |
|---|---|---|---|---|---|---|
| Upper (< 700 m) | 0.7 | 1.7 | 2.9 | 4.0 | 5.1 | 6.4 |
| Mid (700-2000 m) | 1.1 | 2.3 | 3.7 | 4.9 | 5.9 | 6.9 |
| Deep (> 2000 m) | 2.3 | 3.2 | 3.5 | 3.5 | 3.5 | 3.3 |
| Total | 4.1 | 7.2 | 10.1 | 12.5 | 14.4 | 16.6 |

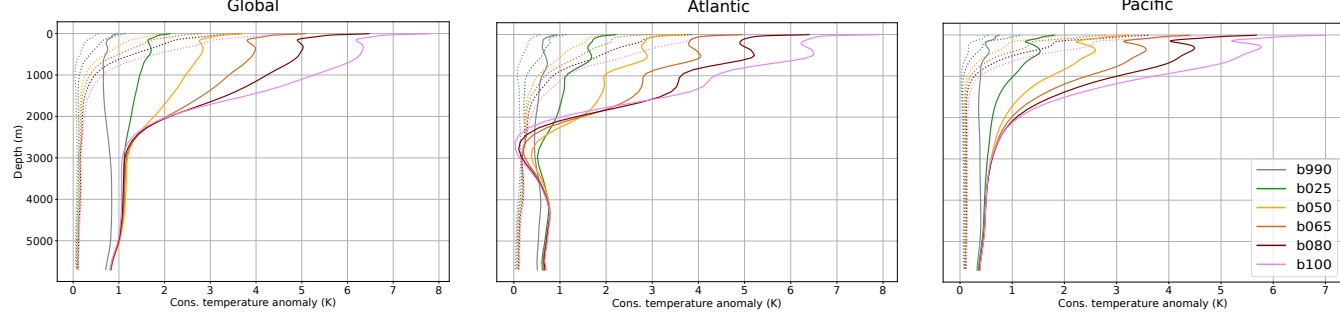

**Figure 10.** Mean profile of the ocean temperature anomaly at depth with respect to the pre-industrial climate for the global ocean (left), Atlantic (middle) and Pacific (right). The dotted lines represent the starting point of the simulations, the solid ones the final state. Shown are 30-yr averages, the conservative temperature variable has been used.

In fact, the distribution of heat in the three layers is also strongly dependent on forcing. In particular, the larger the forcing, the larger the relative fraction of heat that remains in the two upper layers. This is summarized in Table 4, which shows the heat content anomaly of each layer at the end of the run with respect to the pre-industrial climate. Table S1 and Figure S6 in the Supplementary are the analogs of Table 4 and Figure 9, but showing the relative contribution of each layer to the total OHC. When stabilizing at the 1990 forcing, more than half of the heat absorbed by the global oceans is transferred to the deepest

layer in 1000 years. On the opposite, just about 20% of the additional oceanic heat is absorbed by the deep layer at the end of the b100 simulation. Apart from b990, the final heat content in the deep layer is in all cases between 3.2 and $3.5 \times 10^{24}$J, with a maximum for the intermediate forcing of b050 and b065. This despite the fact that, due to the simulation setup, the higher forcing cases had more time to accumulate heat in the deep ocean, since they started from later stages of the SSP5-8.5 simulation.

The final state of the ocean can be appreciated from Figure 10, which shows the mean profile of the ocean temperature anomaly at depth with respect to the pre-industrial climate for the global ocean (left), Atlantic (middle) and Pacific (right), both at the start (dotted) and end (solid) of the simulations. The b990 simulation appears to be quite close to final equilibrium, with an almost uniform distribution of the additional heat along the depth profile. The other simulations are instead increasingly away from a uniform redistribution of heat, in particular below 2000 m, with most of the heat still trapped in the surface layers

for the extreme forcing cases. For comparison, the final equilibrium state of the deep ocean in the abrupt 4xCO$_2$ experiment by Li et al. (2013), showed an almost uniform 8 K temperature anomaly at depth after 5000 years. If the equilibrium state requires a uniform redistribution of heat at depth, at the current pace equilibrating the deep ocean will still take centuries for the low forcing b025 and millennia for b100. However, the final temperature anomaly profile at depth might also be forcing-dependent, as found for example by Rugenstein et al. (2016a), so this may be an over-estimate for lower forcing levels.

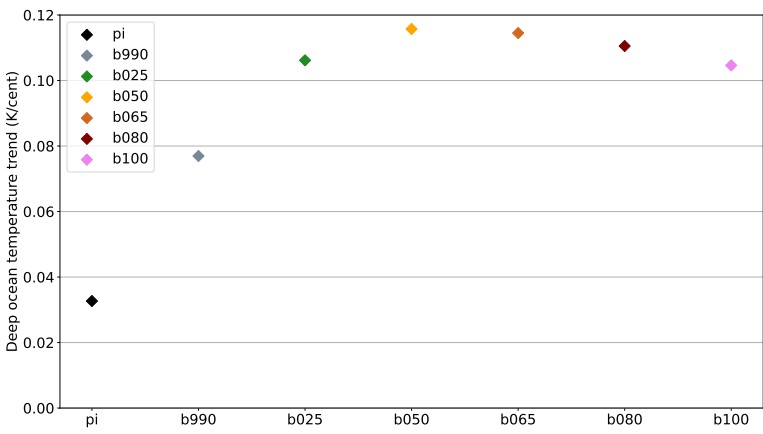

**Figure 11.** Linear trend (K/century) of the deep ocean mean temperature (at depths > 2000 m) in all simulations and in the pre-industrial control.

Interestingly, the Atlantic ocean shows a non-linear behaviour in the region between 2000 and 3500 meters depth, with the highest forcing levels corresponding to the lowest warming. This is not observed for the Pacific ocean, which behaves quite linearly also in this depth region.

The fact that the deep ocean heat uptake is largely independent from the external forcing was an unexpected finding. In particular, one would expect the rate of heat uptake to be larger for larger warming in the surface layers, through the advection of warmer surface waters. However, as shown in Figure 11, the trend in the deep ocean temperature does increase with forcing in b990, b025 and (slightly) b050, but then reaches an absolute maximum and starts to slowly decline up to b100. The drift (positive trend) in the pre-industrial deep ocean is a common feature of CMIP models and is mainly due to computational reasons, since the complete equilibration would require a multi-millennial spin-up simulation. The forced trend in b990 to b100 is however 3 to 4 times larger than that in the pre-industrial, proving that the deep ocean is indeed *feeling* the increased forcing.

The downward transfer of heat in the ocean is the result of the complex interplay between various processes (Gregory, 2000; Hieronymus and Nycander, 2013; Exarchou et al., 2015), namely: advection of heat (including both the resolved advection and the parametrized eddy advection in the model), dia- and isopycnal diffusion, vertical mixing (including wind mixing and convection). In the NEMO model, the pre-industrial equilibrium at depth is maintained by a balance between the geothermal heating at the bottom, the downward heat flux from diapycnal diffusion, and the upward heat flux from advection (resolved + eddy) and isopycnal diffusion (Hieronymus and Nycander, 2013). Interestingly, the resolved advection provides a downward heat flux, but the parametrized eddy-induced advection of heat is upward, resulting in a net upward flux (Hieronymus and Nycander, 2013). It is outside the scope of this paper to provide a complete budget of all the processes involved in the deep ocean warming, which would require either online diagnostics (Hieronymus and Nycander, 2013) or offline calculations to reproduce

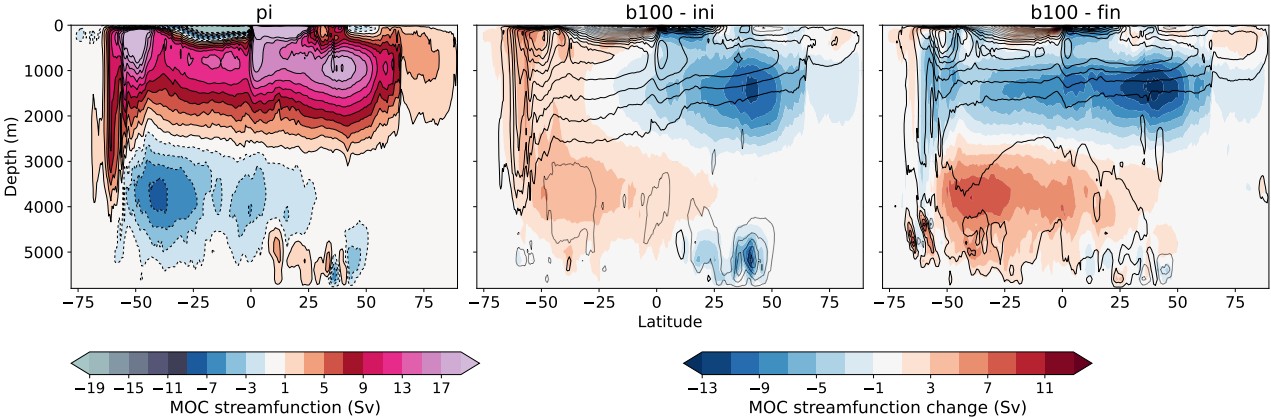

**Figure 12.** Global MOC streamfuction (Sv) for the average pre-industrial (shading, left panel), and the initial and final states of the b100 simulation (mid and right panel; shading: difference to PI, contour: full field at steps of 3 Sv). Separate streamfunctions for the Atlantic and Pacific/Indian oceans can be found in Figures S8 and S9.

the model parametrizations (Exarchou et al., 2015). Nevertheless, we briefly discuss here the role of the heat advection by the resolved circulation.

We hypothesize that the reduced efficiency in the deep ocean heat uptake at larger forcing can be partly explained by a decrease in the downward heat advection to the intermediate and deep ocean layers, due to a reorganization of the global Meridional Overturning Circulation (MOC). Figure 12 shows the streamfunction of the global MOC for the average pre-

industrial (left panel), and the initial and final states of the b100 simulation (mid and right panel). A strong damping of the upper MOC branch in the Northern hemisphere can already be appreciated at the beginning of b100 (Figure 12, middle panel), and is due to the well known reduction in the strength of the Atlantic MOC (AMOC) during SSP5-8.5 (e.g. Bellomo et al., 2021). The circulation in the Southern ocean is still quite unaffected or even strengthened at this stage. Interestingly, during the millennial b100 simulation, the damping of the MOC proceeds mostly at depth, producing a strong reduction of the circulation

below 1000 m depth in the final state, particularly evident in the Southern ocean (Figure 12, right panel).

The changes observed for b100 are also seen in the other simulations, but their amplitude is amplified in the higher forcing cases. To better appreciate the behaviour of the upper and lower branches of the MOC in all simulations, we focus here on two indices, shown in Figure 13:

- the maximum of the Atlantic MOC (AMOC) between 30 and 50 N and 500 and 2000 m (Figure 13, left panel);

- the average value of the abyssal global MOC in the Southern Ocean (SMOC) between 30 and 50 S and 3000 to 4000 m depth (Figure 13, right panel).

The boxplot shows the distribution of the yearly average indices: one for the pre-industrial (black) and two boxes for the first and last 50-year period of each simulation (colors in legend). The AMOC is suppressed at the beginning of the simulations due

to the increased forcing, but then slowly recovers, with a final state which is 1-2 Sv stronger (still weaker than pre-industrial).
The recovery of the AMOC at multi-centennial timescales has already been discussed (Li et al., 2013; Swingedouw et al., 2007; Lembo et al., 2020) and it has been attributed to various negative feedbacks, including increased northward transport of salinity from the tropical Atlantic (Swingedouw et al., 2007). However, the AMOC is not restored to previous conditions: the circulation increases in strength but becomes shallower, being progressively constrained to the upper region of the ocean, as shown by the decreasing depth of the maximum (Figure S5). In fact, the final global-mean ocean state (Figure 10) is less stratified in the upper region (down to about 1000 m) with respect to the beginning, favouring the recovery of the AMOC (Swingedouw et al., 2007), but more stratified below, due to the fact that the deep water has warmed at a much slower pace. Thus, the AMOC is confined upwards and it is less able to transfer heat to intermediate depths.

The abyssal SMOC cell (Figure 13, right panel) experiences an initial reduction, proportional to forcing, and then completely collapses in all simulations apart from b990. This is relevant since the Southern ocean is a key region for deep ocean heat uptake (Morrison et al., 2016; Gjermundsen et al., 2021), and the reduced ventilation at depth prevents warmer waters to reach the deepest layers, thereby damping the dependence on forcing of the response. The collapse of the abyssal SMOC cell is likely connected to the complete disappearance of deep convection at depths larger than 2000 m in all simulations apart from b990 and b025 (also substantially reduced), following the mechanism proposed by (Gjermundsen et al., 2021).

The fact that the reduced downward heat advection is at least partially responsible for the reduced efficiency of the deep ocean heat uptake at higher forcing is further supported by the non-linear behaviour at 2000-3500 meters depth seen in the Atlantic temperature profile (Figure 10, middle panel). Starting from simulation b050 up to b100, it is evident that the temperature anomaly in these layers decreases as the forcing increases. This trend is likely associated with the dynamical adjustments occurring in the Atlantic Meridional Overturning Circulation (AMOC), which tends to become shallower and weaker in the more extreme warming scenarios (see Figures 12, 13). Changes in other processes might also contribute to the heat budget. For example, Morrison et al. (2016) observe that a decrease in the along-isopycnal temperature gradient in warmer climates would reduce the upward heat flux due to eddies and diffusion, thereby increasing the warming at depth. The current understanding of vertical heat fluxes in the deep ocean is still partial and the response is strongly model-dependent, governed by the specific implementation of eddy and diffusion parametrizations (Exarchou et al., 2015) or by the strength of the deep convection in the model Gjermundsen et al. (2021).

## 5    Conclusions

We presented here a set of abrupt stabilization simulations performed with EC-Earth3 and analysed the equilibration pathway of the climate system with a focus on the multi-centennial timescales and on the dependence on different levels of forcing. The setup is compatible and to some extent inspired by the LongRunMIP ensemble. The six simulations - with external forcing ranging from that of year 1990 (b990) to year 2100 of the SSP5-8.5 scenario (b100) - show a final temperature increase between 1.4 and 9.6 K with respect to pre-industrial. The global GTAS stabilizes in all simulations after 600 to 900 years, apart from b050 which still shows a significant residual trend at the end of the simulation. The global mean hydrological cycle response

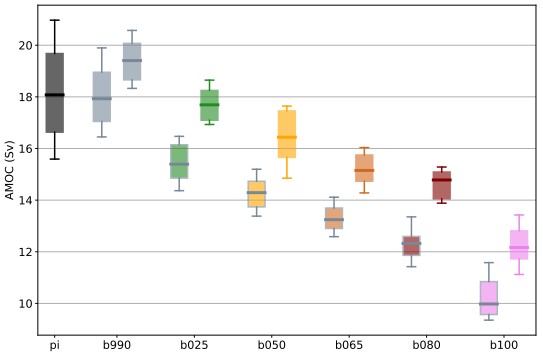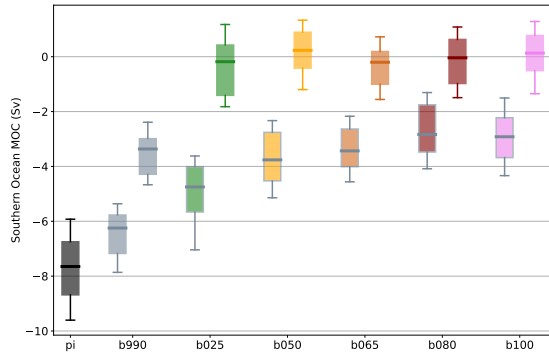

**Figure 13.** Boxplot of the yearly distribution of AMOC (left panel) and SMOC (right panel) strength at the beginning and end of each simulation (50 yrs).

intensifies during stabilization, leading to an increased trend per degree of global warming, due to the increased surface energy available for evaporation (Wu et al., 2010). Winter Arctic sea-ice collapses in two simulations (b080, b100), consistent with Meccia et al. (2020), and has apparently reached a tipping point in b065, where the system switches periodically between a low and a high sea-ice cover state.

Analyzing the behaviour of the climate feedback parameter across the different simulations, we find that:

- the feedback parameter tends to become less negative in the second half of the simulation, in agreement with the results by Rugenstein et al. (2020);

- the climate feedback is more negative (larger in magnitude) for increasing forcing/warming. The EC-Earth3 model seems to be among the few models showing this behaviour, while most models show a reduction of the feedback magnitude for larger forcing, as reported by Bloch-Johnson et al. (2021).

However, caution is needed when making direct comparisons of these results with Bloch-Johnson et al. (2021) and other studies utilizing abrupt NxCO2 simulations (e.g. Meraner et al., 2013), as there are differences in our experimental setup, encompassing both the initialization process and the external forcing components (such as other greenhouse gases and aerosols). Further investigations are necessary to explore potential physical mechanisms that can explain this behavior, including the role of different forcing agents.

Following the global overview, an analysis of the temperature and precipitation patterns during the stabilization phase (Section 3.3) was undertaken, with a focus on comparing the low forcing (b025) and high forcing (b100) simulations. In all simulations, there is a drastic acceleration of the warming in the Southern Ocean, in agreement with previous studies (Andrews et al., 2015; Rugenstein et al., 2020). The pattern of the response is generally robust across the simulations, but shows some dependency on forcing/warming. In particular, at lower forcing the increase is concentrated to the South of the Atlantic and

Indian oceans, and extends to the Pacific region only at higher forcing (Figure 7). The precipitation patterns also evolve during stabilization. In general, drying trends found in the sub-tropical oceans and in Mediterranean-like hotspots during SSP5-8.5 tend to reduce during stabilization, or even to reverse. This is generally in line with Zappa et al. (2020), although the inversion of the trend is more clear for larger forcing, especially for the southern subtropics. Even in the regions where the trend inverts during stabilization, the total precipitation change at the end of the simulation with respect of pre-industrial is generally in the same direction as the transient one, but reduced in amplitude. Also, at large forcing (b100), a strong perturbation is observed in the tropical region, with drying over the West Pacific warm pool and wettening of the Amazon, hinting at significant changes in the Walker circulation, likely linked to the change in the SST warming pattern (Chadwick et al., 2013).

The multi-centennial timescales of the climate response are driven by the transfer of heat to the deep ocean (e.g. Armour et al., 2016). We analyzed the heat stored in the upper (0-700 m), mid (700-2000 m) and deep layers of the global ocean in the simulations, and found that the distribution of the heat absorbed in the three layers is strongly dependent on forcing: the heat is almost uniformly distributed along the depth profile in lowest forcing (b990), while in the largest forcing the upper ocean absorbed most of the additional heat. We found that the rate of deep ocean warming is almost independent from the external forcing and the maximum is reached at an intermediate forcing level (b050), meaning that the efficiency of the heat transfer to the deep ocean decreases for larger forcing and at longer timescales. Rugenstein et al. (2016b) also reported the non-linear behaviour of the deep ocean heat uptake at these timescales across different levels of forcing, studying a set of EMIC simulations. We hypothesize that this is due to a decreased ventilation of the deep ocean with larger forcing, due to the ocean dynamical adjustments, implying a general reorganization of the MOC. Indeed, the AMOC slightly recovers in intensity during the simulation, but the AMOC cell is increasingly shallower at larger forcing and less able to transport heat at depth. The non-linear behaviour observed for the warming at 2000-3500 meters depth in the Atlantic - with decreasing anomalies for larger forcing - further supports this hypothesis. At the same time, the abyssal MOC in the Southern ocean is first damped proportionally to forcing and then completely collapses during stabilization, further reducing the mass flux to the deeper layers.

The results highlight the importance of studying multi-centennial timescales of climate change to better understand and constrain processes that will play a role in determining the final response of the climate system once GHGs concentrations are stabilized. Many questions remain open and require further investigation. Specifically, the dynamics of deep ocean heat uptake poses a significant puzzle, as our findings indicate that the rate of deep ocean warming remains nearly constant regardless of external forcing, after peaking at an intermediate level. While we highlight here the significance of oceanic dynamical adjustments, further exploration is needed for other mechanisms, particularly focusing on parametrized heat diffusion and turbulent fluxes. A potential avenue forward entails investigating the sensitivity of these mechanisms to model resolution and tuning, which could help in understanding inter-model variations and bridging the gap with lower complexity models. Additionally, the role of deep ocean heat uptake in shaping the surface response, the evolution of climate feedbacks and the stabilization timescales is also crucial and deserves further investigation.

*Data availability.* The data of the simulations are available upon request to the corresponding author.

*Author contributions.*  FF, PD, SC and GZ planned the simulation setup, PD and FF ported the model code on the HPC machine, FF performed most of the simulations, managed the data produced, and performed most of the data analysis, with feedbacks from all authors. VM run one of the simulations and helped with the AMOC analysis. FF wrote the first draft of the paper; VM, PD, GZ, AB, KB, VL and SC all commented and/or edited the manuscript.

*Competing interests.*  The authors declare no competing interest.

*Acknowledgements.*  The simulations have been performed under the project QUECLIM (Quasi-equilibrium climates at fixed external forcing) at CINECA (ISCRA project IscrB_QUECLIM) and ECMWF (special project spitfab2), which we acknowledge for support and computing resources. KB has received funding from the European Union's Horizon 2020 research and innovation programme under the Marie Sklodowska-Curie grant agreement No. 101026907 (CliMOC). This work has received funding from the Italian Ministry of Education, University and Research (MIUR) through the JPI Oceans and JPI Climate "Next Generation Climate Science in Europe for Oceans" -
ROADMAP Project (D. M. 593/2016) (AB, VL, SC), and from the European Union's Horizon 2020 research and innovation program under grant agreement No. 820970 (TiPES) (VLM, VL, SC). This is TiPES contribution #259. VL, SC, VLM and GZ have been supported by the European Union's Horizon Europe research and innovation program under grant agreement no. 101081193 (OptimESM project).

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
