# Peer review of "Multi-centennial evolution of the climate response and deep ocean heat uptake in a set of abrupt stabilization scenarios with EC-Earth3"

_Earth System Dynamics, 2023_

## Author Comment (AC1)

**Reviewer 1**

It is important and interesting to conduct a set of well-designed 1000-year-long abrupt stabilization simulations to understand the final-state response of global climate to various stabilized external forcing levels. In this study, the EC-Earth3 model estimates a range of 1.4 K to 9.6 K global mean surface temperature (GMST) increase relative to preindustrial level if the external forcing is suddenly stabilized at specific levels in historical (1990) and SSP5-8.5 (2100). The evolution of the pattern of surface warming, and precipitation change in the long simulations clearly reveal the effect of the deep ocean heat storage and its feedback to global and regional surface climate at a long time scale. The results are well presented and advance our understanding of the equilibrium response of global climate to different levels of anthropogenic forcing. Some concerns are the interpretation of the underlying mechanism for deep ocean warming and its non-linear response to different magnitudes of external forcing.

> We thank the reviewer for the comments and suggestions. Replies to specific points are below.

Minor comments:

(1) Lines 5-7, 141-142: The author may need to compare the climate sensitivity of EC-Earth3 with other CMIP6 models when estimating the GMST response and associated Paris Agreement target.

> We will add a comment comparing the climate sensitivity of EC-Earth3 with other models' results.

(2) Lines 38-39, the upper ocean fast warming and the slow but persistent deep ocean warming are well noted in the results from the experimental and diagnostical approach of Held et al. (2010) and Long et al. (2014). The deep ocean feedback on the surface warming pattern and precipitation change is also discussed in Zappa (2020), King et al. (2020), Kim et al. (2022), etc.

> Thank you for your suggestions, we will better link our results to these references in the revised version.

(3) Figure 7: the shift of the precipitation trend (dry to wet; wet to dry) is mainly distributed at the boundary of the regions with negative and positive precipitation trends, which may not be significant. The role of internal variability can be large during stabilization simulation, especially for precipitation change. The author may pay attention to these issues and related conclusions. However, it is worthy to discuss the precipitation pattern with significant trends during the stabilized period following the mechanisms proposed by Chadwick et al. (2013a, b).

We agree that the internal variability may be large when looking at precipitation trends. To reduce the impact of internal variability, the maps in Figure 7 are computed as a regression of local precipitation over low-pass filtered GTAS. Many parts of the world indeed show non-significant trends, but some key regions do reach the 5% significance level. The assessment of the trend inversion is complicated in some cases since either the transient or the stabilization simulations may show non-significant trends in those regions. However, some patterns are clear. For example, in Northern subtropics where intense drying is observed during the transient, the trend during stabilization is either close to zero and non-significant (Mediterranean), or significantly positive (central America). For the Mediterranean, clearer trends are indeed observed "at the boundaries", but this is not necessarily the case for other regions. We will briefly discuss this issue in the revised manuscript.

(4) Lines 260-262: This fact can actually be explained by the fast upper ocean and slow deep ocean response to increasing external forcing. As long as the GHG forcing is gradually increased in SSP5-8.5, the upper ocean warms much faster than the deep ocean and would accumulate more heat in the upper ocean, leaving a relatively small fraction of deep ocean heat storage.

We have not fully understood the reviewer's comment. Those lines read "Apart from b990, the final heat content in the deep layer is in all cases between 3.2 and 3.5×$10^{24}$ J, with a maximum for the intermediate forcing of b050 and b065. This despite the fact that, due to the simulation setup, the higher forcing cases had more time to accumulate heat in the deep ocean, since they started from later stages of the SSP5-8.5 simulation". The upper ocean keeps storing heat during the whole stabilization simulation, though at a decreasing pace. The upper ocean heat uptake rate is proportional to the forcing, as expected. What we find curious is, instead, that the deep ocean heat uptake rate changes only slightly with the forcing and in a non-monotonic way.

(5) Lines 320-328, 348-352: The deep ocean warming in the Indo-Pacific Ocean is also a result of the heaving effect of the inter-basin water redistribution due to AMOC weakening, which is evident maximizes at 300-3000m layer (Sun et al. 2022). The underlying mechanisms of deep ocean warming may differ substantially between the Indo-Pacific and Atlantic Oceans. It is necessary to show the response of Indo-Pacific and Atlantic Ocean temperature and meridional overturning circulation, respectively, in Figs. 9 and 11. The dynamical effect of the ocean circulation change and non-linear recovery of AMOC may also be important in explaining global deep ocean temperature change and hence OHC change under different levels of stabilized external forcing.

Thank you for the suggestion. We will compute and show the meridional streamfunction and the temperature profile also for the Atlantic and Indo-Pacific oceans separately.

Refs:

Chadwick, R., I. Boutle, and G. Martin, 2013a: Spatial patterns of precipitation change in CMIP5: Why the rich do not get richer in the tropics. J. Climate, 26, 3803–3822.

Chadwick, R., P. L. Wu, P. Good, and T. Andrews, 2013b: Asymmetries in tropical rainfall and circulation patterns in idealised CO2 removal experiments. Climate Dyn., 40, 295–316.

Held, I. M. et al. Probing the fast and slow components of global warming by returning abruptly to preindustrial forcing. J. Clim. 23, 2418–2427 (2010).

Long, S.-M., S.-P. Xie, X.-T. Zheng, and Q., Liu, 2014: Fast and slow responses to global warming: Sea surface temperature and precipitation patterns. J. Climate, 27, 285–299.

Kim, SK., Shin, J., An, SI. et al. Widespread irreversible changes in surface temperature and precipitation in response to CO2 forcing. Nat. Clim. Chang. 12, 834–840 (2022). https://doi.org/10.1038/s41558-022-01452-z

King, A.D., Lane, T.P., Henley, B.J. et al. Global and regional impacts differ between transient and equilibrium warmer worlds. Nat. Clim. Chang. 10, 42–47 (2020). https://doi.org/10.1038/s41558-019-0658-7

Sun, S., Thompson, A. F., Xie, S. P., & Long, S. M. (2022). Indo-Pacific warming induced by a weakening of the Atlantic meridional overturning circulation. *Journal of Climate*, *35*(2), 815-832.

Zappa, G., Ceppi, P., and Shepherd, T. G.: Time-Evolving Sea-Surface Warming Patterns Modulate the Climate Change Response of Subtropical Precipitation over Land, Proceedings of the National Academy of Sciences, 117, 4539–4545, https://doi.org/10.1073/pnas.1911015117, 2020.

**Reviewer 2**

Review of "Multi-centennial evolution of the climate response and deep ocean heat uptake in a set of abrupt stabilization scenarios with EC-Earth3" by Federico Fabiano and co-authors

The paper surveys the response of one model to a range of stabilization scenarios sampling many forcing levels of $CO_2$. The authors discuss the response of surface warming patterns, precipitation, sea ice and deep ocean heat storage in the different scenarios. The text is well written, easy to follow, and the figures are clear and convey the points the authors make in the text.

My big issue with this paper is that there is no hypothesis, one single very general and in a sense irrelevant question (which is not answered beyond what is known about it already), no increased understanding of a concrete physical process, and a poor embedment in the literature for some of the metrics' surveyed. It is merely *reported* or "explored" how one model responses to a range of forcing scenarios. "What happens in this model?" is not a scientific question.

The subject itself is interesting, not too well studied and does actually have open scientific questions. Hence, instead of merely surveying all sorts of "measures", I suggest the authors develop 1-3 specific questions and hypotheses (from the current understanding of these timescales presented in the literature). I point out below which questions I think merit attention but these are only suggestions to illustrate where my thinking stems from.

> We thank the reviewer for this comment and accept the criticism, which we will tackle in the revised manuscript. We agree that there is the need to better address our objectives and scientific questions in the introduction. Of course, "what happens in this model?" is not our scientific question. Our goal is to investigate climate stabilization pathways on multi-centennial timescales under different levels of forcing/warming, something that has never been explored before with a full GCM, and can complement what has been done so far by LongRunMIP. In this regard, LongRunMIP has been our main reference in the inspiration and setup of this work, and we will better reference it and related recent results in the introduction and discussion sections.

> Given the effort and size of the computing project undertaken, we first aim to provide here a complete assessment of the simulations, presenting the setup and a general overview. Furthermore, we select a few scientific questions to be investigated in depth, which are:

- How do temperature and precipitation patterns evolve during multi-centennial timescales in EC-Earth3? And how does this depend on the level of forcing/warming?
- What is the response of the deep ocean to the sudden stabilization of external forcing? How does this response evolve at multi-centennial scales and how does it depend on different levels of forcing?

Besides, following the reviewer's suggestions, we will add to the first section (overview) a focus paragraph about:

- How do climate feedbacks depend on the strength of the external forcing and on the mean state change over multi-centennial timescales?

We will better address our scientific questions in the introduction and throughout the revised version of the manuscript.

Minor and major comments are mixed below.

Line 9 "the most striking feature being a drastic acceleration to the warming in the Southern Ocean" – the fact that the Southern Ocean lags warming is decades old and definitively not surprising or worth reporting. See e.g. review in Armour et al. 2016. This paper does not add any new knowledge to the subject.

We are aware that the lag in the Southern Ocean warming is a known result in literature, including Armour et al. 2016, which is indeed already cited at lines 209-210. We will rephrase the sentence in the revised manuscript, extending the references to other existing literature. However, we disagree with the reviewer that this paper does not add any new knowledge to existing literature on this specific aspect, for two reasons:

- This is the first time such long-term warming trends are documented for the EC-Earth3 model, and we present here an important assessment of the model response at long timescales which is certainly fruitful for the entire EC-Earth community and, also, for the broader climate modelling community;
- To our knowledge, the dependence of the multi-centennial warming patterns under different levels of forcing/warming has not been previously studied with a full complexity GCM. The present work includes 6 different stabilization scenarios and it is performed with a GCM characterized by one of the highest resolutions

among the CMIP6 ensemble. In the past, Rugenstein et al. (2016) tackled this topic with an EMIC, while Rugenstein et al. (2020) studied the multi-model ensemble response in LongRunMIP, but did not account for the impact of different levels of radiative forcings (e.g. in Figure 3). Indeed, at the end of their Section 2, Rugenstein et al. (2020) reads: "*As not all models contributed several forcing levels, we focus in the following on robust pattern changes in surface temperatures and feedbacks, which occur in most or all simulations irrespective of their overall temperature anomaly or forcing level.*"

We will expand the discussion regarding this issue (giving more credits to relevant literature) and better clarify in the introduction/discussion our focus on the dependence of the multi-centennial response on different levels of forcing.

p.2 "One fundamental question remains unanswered: which will be the equilibrium state of the climate once all of the warming linked to a specific level of forcing is realized? " – so what is the answer? What do you show which goes beyond Li et al. 2013, Rugenstein et al. 2019, Mitevski et al. 2021? How does this equilibrium state relate to the policy implication the study is motivated with? Why does this state matter? Has the real climate system ever been in that state or is it expected to move there (I don't think so)? What is "the state of the climate"? Why is are the spatial patterns relevant? How far into the final equilibration are your simulations in the year 3000?

We agree with the reviewer that the question was formulated in a somewhat naive way. Of course, we do not have a final answer to the question and do not claim to do so, nor can we reply to all the above questions. However, we are firmly convinced that some of the analyses presented here are original and may help to get closer to the answer to the above general question. To give you some example associated with the above-cited literature:

● The study of Li et al. (2013) consists of a single model study with ECHAM5/MPIOM, which is different from EC-Earth3, and it is characterized by an older configuration at lower resolution. So, results from a more recent state-of-the-art model, regardless of the agreement (or not), are worth to be presented and discussed.
● The multi-model ensemble analyzed by Rugenstein et al. (2019, 2020) does not include EC-Earth3. Moreover, as mentioned above, there is no focus in Rugenstein et al. (2019, 2020) on the impact of different levels of forcing, as it is done here. Indeed, Rugenstein et al. (2020) gave the inspiration to tackle the question in this work.

- All the simulations in Mitevski et al. (2021) last only 150 years and therefore they do not take into account the multi-centennial scales considered in our study.

Of course, as the reviewer correctly pointed out, several aspects remain unsolved and we hope to be able to address this in future works: for instance, the policy implication will be further explored in a follow-up study that will focus on climate impacts, also comparing to existing 1.5° and 2°C stabilization scenarios.

About the equilibrium state. The real climate has never been in such a stable state persistently, but it has been in a state with no forcing (and impacts) due to human activity. In a time window where we can consider constant orbital parameters, we believe that the question of "what is an equilibrium state and how long does it take for the system to realize all the warming implied by a specific anthropogenic forcing" is a relevant question for the long-term future (thousand years forward). Here we only estimate a stabilization time for the global surface climate. However, the analysis could be extended in order to consider specific regions. For the deep ocean, we are able to estimate a stabilization time only under the hypothesis that the additional warming at equilibrium will be close to uniform along the water column (as found by Li et al., 2013).

Computational costs are mentioned as the reason for few equilibrated simulations. First of all there are not so few around by now, and second, this argument was true ten years ago, but not anymore today. There are km-scale simulations by now which used order of magnitude more computational resources than argued about there, initial conditions ensembles with tenths of thousands of years, perturbed parameter simulations with several thousand years. Equilibrating simulations might not have the strongest lobby, but it is not correct anymore that the computational resources are not existing. Somewhat related, Li et al. 2013 and Zickfeld et al. 2013 are mentioned but these papers are ten years old and the discussion has moved on. What do modern EMICs say about the issues? Are they even still around? What features of the equilibration or the equilibrium do EMICs and GCMs share and where do they differ and how? What do these differences tell us about the real world? Do we trust GCMs more just because they also have clouds, which we know are extremely parameterized?

We thank the reviewer for the comment and the interesting point about the difference between GCMs and EMICs. We acknowledge that many of the works on these timescales (for example those cited in Rugenstein et al., 2019) are not properly referenced in the current introduction. We will add a discussion and references to relevant papers in the revised manuscript.

As for the use of EMICs in place of GCMs, we agree that it is interesting to explore these timescales through simplified models and to compare their response to that of GCMs. GCMs are generally better skilled at reproducing the observed climate, since they contain a more reliable representation of physical processes, and thus are more trustworthy when studying climate feedbacks and heat storage on long timescales. This is the case for example when studying deep ocean circulation and heat uptake, which are not properly represented in EMICs. Nevertheless, despite the undisputable incremental gain in computational resources, there are some aspects of the climate response requiring an investigation that is hardly reachable with coupled GCMs at CMIP-class resolution. Given this, besides studying the few affordable millennial GCM simulations (ad done in LongRunMIP), it is beneficial to resort to simplified models as a step in the ladder of model hierarchies, coherently with the protocol envisaged by Held in 2005 (Held 2005, Jeevanjee et al. 2017).

Held, Isaac M. "The gap between simulation and understanding in climate modeling." Bulletin of the American Meteorological Society 86, no. 11 (2005): 1609-1614

Jeevanjee, N., Hassanzadeh, P., Hill, S., and Sheshadri, A. (2017), A perspective on climate model hierarchies, J. Adv. Model. Earth Syst., 9, 1760– 1771, doi:10.1002/2017MS001038.

Following, the LongRunMIP project is mentioned once but not at all discussed. The project has 14 or so models, some of which have three forcing levels. The literature discusses things like equilibration timescales of the surface warming pattern, the deep ocean, the Atlantic meridional overturning circulation, top of the atmosphere radiative imbalance, polar amplification, ENSO and the temperature dependence of feedbacks. Where do the findings discussed here go beyond that? Other recent effort include Dunne et al. 2020 who gathers simulations longer than 800 years and some models CMIP6 extended their required protocol to well beyond 150 years of step forcings (see e.g. list in Bloch-Johnson et al. 2021). It is argued that most of these simulations are step forcings. However, they find exactly the same things as pointed out here. Until when does the scenario before the stabilization matter? Does it matter at all? For what?

We agree with the reviewer that the LongRunMIP project is relevant and we will better reference related recent results in the introduction. Overall, we admit that our introductory section did not sufficiently highlight the important existing literature. However, the findings discussed in this paper are not directly comparable (they study different aspects of climate change) and "go beyond" the LongRunMIP results in various respects:

- the EC-Earth3 model has never been run on these timescales, and we consider the analysis of its long-term response under various forcing relevant for publication, as had been the case for other single model results in literature;
- we focus here not only on the climate response at multi-centennial timescales, but also on its dependence across various levels of forcing and mean states (6 forcing levels/stabilization pathways), which is not treated in any LongRunMIP paper, as far as we know, nor elsewhere;
- we believe that the rate of deep ocean heat uptake and its dependence on forcing at these long timescales is a topic that has not yet been studied much and still presents many open questions. We believe that Section 4 contains original contributions to the topic, which are not treated in the papers suggested by the reviewer, nor elsewhere (to our knowledge).

Page 6 line 150, stabilization – more relevant than surface temperature would be the top of the atmosphere energy imbalance or the surface flux imbalance or the accumulated ocean heat content. How can the surface temperature be equilibrated, while the TOA is not (as can be seen in 4 but would be more obvious in a timeseries)? Through which processes do the deep ocean and TOA communicate at these timescales? Is the connection between deep ocean, surface, and top of the atmosphere forcing dependent?

The equilibrium of the surface layer is not only governed by the TOA, but by the balance between the TOA and the rate of ocean heat uptake. In this sense, it is perfectly coherent that the surface temperature stabilizes even if the TOA is not zero, since the incoming energy is completely absorbed by the oceans. As for the forcing dependence, the non-linear behaviour observed hints at some forcing dependence of the processes involved, the most obvious candidate being the large scale ocean circulation.

By looking at the residual trend in GTAS, we aimed here at defining an objective criterion to define the beginning of what Li et al. (2013) call "quasi-equilibrium", or a period during which the change in GTAS is very small, but the ocean heat uptake goes on (and deeper layers show progressively larger trends). The analysis shows that indeed the simulations that reach this quasi-equilibrium are those that show a smaller energy imbalance at TOA at the end. But defining this through the TOA would require the choice of an arbitrary threshold, while using the significance of the trend in GTAS allows to avoid this.

Sea ice/Fig.3 are actually one of the least explored issues in my understanding of the literature. Open research questions could be whether you can predict the forcing level at which sea ice will collapse or stay below a certain threshold? What sets the rate of decay of sea ice – the global

warming, local warming? How relevant is the sea ice response to the non-linear behavior of the ocean heat uptake? How unrealistic is this response shown here given the fixed ice sheets? Is this time evolution dependent on the rather large mean-state bias? Sea ice feedback itself is pretty dependent on that (e.g., Kajtar et al. 2021).

> We absolutely agree with the reviewer on this topic. Some of the authors of the current manuscript have already worked on the topic and we are looking forward to performing some analysis in this direction. However, given the relevance and complex discussion that such a topic might require, we prefer to avoid discussing the complex non-linearity here.

An interesting unexplored question around Fig.4 would be what causes radiative feedbacks (the slope of the lines) to be more negative with increasing forcing levels? See Jonah Bloch-Johnson et al. 2021 who finds that most models do the opposite than your model (increasing feedbacks with increase forcing, although this seems to be extremely model dependent) and Mitevski et al. 2021 for this being discussed earlier and pointing out the open questions around feedback temperature and feedback forcing dependence.

> We agree that the feedback dependence on forcing and mean state is an interesting behaviour observed here. We already did some analysis in that respect, but we originally planned to leave it for a separate publication, since further study was needed. However, following the reviewer's suggestion, we plan to add a brief focus on this in the revised manuscript, possibly including a new figure. We thank the reviewer for the relevant literature suggestions.

Is b100 actually stabilizing at -0.2Wm^-2?, Fig.4 doesn't look like it?

> The spurious energetic imbalance of the EC-Earth3 model is around 0.2 W/m2 (energy is spuriously created). This can be estimated by subtracting the TOA minus the surface fluxes. We checked the balance between TOA and surface net heat fluxes also in the stabilization simulations, and verified that the spurious energy source is not significantly forcing- or state-dependent. We thus expect the TOA net flux to stabilize around -0.2 W/m2. The net TOA at the end of the b100 run is about -0.12 W/m2 (Table 2), so we expect further decrease.

Page 10 line 200 "This is well known… due to larger thermal inertia…" Mike Byrne has a range of papers showing that this is actually not the correct explanation for the land-ocean contrast. If the thermal inertia argument was true, in the equilibrium ocean and land should have the same warming – which they don't suggested by Fig.5 and even after 4000 years they are not

(LongRunMIP). What processes set the land-ocean heating contrast equilibration? When does the contrast equilibrate?

> We think there is a possible misunderstanding on this point. When dealing with the land-sea thermal contrast, two problems are relevant: 1. the characteristic timescales of land and ocean warming; 2. the equilibrium land-sea contrast. The sentence at line 200 actually refers to the first problem and the large thermal inertia of the oceans (together with the efficient transfer of heat at depth) is indeed a relevant factor in determining the longer timescale of ocean warming (see e.g. Joshi et al., 2007; Boé and Terray, 2014). Regarding the second problem, we agree with the reviewer that several mechanisms come into play (e.g. lapse-rate, humidity, circulation, ..) and the thermal inertia is not the main factor. We will rephrase the sentence to avoid this confusion and briefly expand the discussion on the point in the revised manuscript.

> Joshi, Manoj M., Jonathan M. Gregory, Mark J. Webb, David M. H. Sexton, and Tim C. Johns. "Mechanisms for the Land/Sea Warming Contrast Exhibited by Simulations of Climate Change." *Climate Dynamics* 30, no. 5 (April 1, 2008): 455–65. https://doi.org/10.1007/s00382-007-0306-1.

> Boé, Julien, and Laurent Terray. "Land–Sea Contrast, Soil-Atmosphere and Cloud-Temperature Interactions: Interplays and Roles in Future Summer European Climate Change." *Climate Dynamics* 42, no. 3 (February 1, 2014): 683–99. https://doi.org/10.1007/s00382-013-1868-8.

> Byrne, Michael P., and Paul A. O'Gorman. "Land–Ocean Warming Contrast over a Wide Range of Climates: Convective Quasi-Equilibrium Theory and Idealized Simulations." *Journal of Climate* 26, no. 12 (June 15, 2013): 4000–4016. https://doi.org/10.1175/JCLI-D-12-00262.1.

Section 4 The forcing dependence of the overturning circulation has been discussed a while ago by Rugenstein et al. 2016 (with the same findings as here, except that the Southern Ocean overturning played a larger role) and more recently by Mitevski et al. 2021. These papers discuss the non-linear dependence of the ocean heat uptake to forcing and that warming in the equilibrium is not homogeneously distributed. However, there are open questions around this: How model-dependent is this non-linearity? What sets its dependence? Does is matter for more practical issues like the end-of-the-21[st]-century temperatures? Can we learn something about tuning parameters, like diffusivity or vertical mixing, from this behavior? Does the type of forcing matter, i.e. are the response to aerosol and $CO_2$ forcing additive when it comes to ocean

heat uptake? Practically, how do we include this effect when estimating climate sensitivity for example from the last glacial maximum or very warm periods in the past?

> These questions are extremely interesting but, due to experimental setup, we cannot pursue all these aspects, mostly because our modeling setup includes a single model and we decided - as will be better clarified in the introduction - to focus on multiple levels of forcing/warming rather than playing with different modeling setups. The role of model tuning, which is something our group is already working on, will be investigated in a follow up study.

I don't ask for all these questions to be answered in a new version of a manuscript. But developing a few questions and according hypotheses well should result in a paper which goes beyond reporting.

Armour et al. 2016 Southern Ocean warming delayed by circumpolar upwelling and equatorward transport

Dunne et al. 2020 Comparison of Equilibrium Climate Sensitivity Estimates From Slab Ocean, 150-Year, and Longer Simulations

Rugenstein et al 2016 Nonlinearities in patterns of long-term ocean warming

Mitevski et al. 2021 Non-Monotonic Response of the Climate System to Abrupt $CO_2$ Forcing

Bloch-Johnson et al. 2021 Climate Sensitivity Increases Under Higher $CO_2$ Levels Due to Feedback Temperature Dependence

Kajtar et al. 2021 CMIP5 Intermodel Relationships in the Baseline Southern Ocean Climate System and With Future Projections

---

## Author Response (AR2)

Reviewer 1

The authors have carefully addressed most of the concerns and comments I proposed in the previous review report. Only a few issues remained for further revision and clarification.

1. Correct typos like Line 428 Fig. S4 (should be Fig. S6).

Thanks for spotting this, it has been corrected.

2. Add the percentage of the accumulated heat in each layer relative to the total amount in Table 4 like 0.7 (17%) and the author will note that despite the absolute value of the amount of deeper layer accumulated heat remaining nearly constant (ranging around $3*10^{24}J$), the percentage of the heat accumulated in the deeper layer (below 2000m) generally decreases from b990 to b100. This is because in a scenario with a higher forcing (like b100) than b990, the fast upper ocean warming and ocean dynamic adjustment associated with AMOC weakening would primarily cause a larger warming in the upper 2000m but not below 2000m (diffusion is rather slow in transfer heat downward). Therefore, the percentage of the heat stored in the deeper layer decreases from b990 to b100. This is not related to the fact that the higher forcing cases had more time to accumulate heat in the deep ocean, which may be valid for the Indo-Pacific but not the Atlantic. The middle panel in Fig. 10 clearly illustrates this issue that between 2000-3000m, the b100 simulation displays a weaker warming than b990, this is linked to the AMOC-related ocean interior adjustment. The differences between the Indo-Pacific and Atlantic deeper layer response may explain this issue.

We tried to add the information on the relative amount of heat in Table 4, but that makes it difficult to read. We added Table S1 to the Supplementary, above Figure S6, reporting the relative values. A reference to the new table has been added to the text. We agree with the reviewer that the dynamical changes in the AMOC are a possible explanation; this was already stated at lines 489-493 and 541-546 (new version), which have now been rearranged to better make the point.

3. It may not be proper to name it "warming hole" in Line 500, which indeed indicates the role of opposite stratification change above and below 2000m that may also relate to AMOC change. This is an interesting result and may be worthy of more in-depth discussion.

We removed the name "warming hole" which indeed could have been misleading, and substituted that with "non-linear behaviour". We added a sentence to stress the link of this phenomenon with the ocean dynamical adjustments at lines 491-493.

Reviewer 2

Review of "Multi-centennial evolution of the climate response and deep ocean heat uptake in a set of abrupt stabilization scenarios with EC-Earth3"

By Fabiano et al.,

Recommend: minor revision

General comment:
This study presents a thorough analysis of 1000-year abrupt stabilization simulations with EC-Earth3, offering some insights into long-term committed climate change. The simulations, spanning historical and SSP5-8.5 scenarios, reveal temperature increases exceeding Paris Agreement goals. Notably, only the 1990 simulation achieves stabilization below 1.5 degrees warming. The study emphasizes the importance of multi-centennial timescales, noting a decrease in climate feedback parameter magnitude and revealing variations in surface warming patterns. Precipitation changes, particularly in sub-tropical oceans and Mediterranean hotspots, highlight dynamic climate processes. The focus on deep ocean heat storage underscores its role in determining the final state of the climate system. Overall, the findings contribute to our understanding of climate dynamics and have implications for informing climate policies. Although long simulations with coupled climate models have been conducted and presented in many previous studies, the simulations from the present study have much higher resolutions and consider different warming scenarios. The paper exhibits a well-organized structure and skillful writing, with a coherent storyline that enhances the research's accessibility. Overall, I find the paper intriguing and recommend its publication after the necessary revisions.
I have also read previous review comments and the point-to-point reply to all review comments. In the previous review, the comprehensive comments provided by both reviewers have proven invaluable in refining and improving the manuscript during the revision process.

I have a few minor points that require further attention and revision:
1. One main concern is the forcing dependence of the climate feedback parameter: I still do not understand why the climate feedback parameter turns out to be more negative while increasing external forcing. This is different from the previous study of Jonah Bloch-Johnson et al. (2021) and Meraner et al. (2013). I am sure there are also other studies on state-dependent climate feedback and climate sensitivity. The authors have added one more section to discuss the forcing dependence of the climate feedback parameter. But I still cannot an answer in the manuscript or in the reply why the climate feedbacks behave like that. I am sure many things should be done and can be done in further studies on this issue, but it would be great if the authors would provide some explanations. A reasonable physical mechanism will be very important here since the authors argue this is one of the main novelty of the present research.

Meraner, K.⋆ , T. Mauritsen, and A. Voigt, 2013: Robust increase in equilibrium climate sensitivity under global warming, Geophysical Research Letters, 40, 5944–5948.

Thanks for the comment. We first point out that Figure 5 has been updated since the previous version did not include the correction of the TOA fluxes for the model imbalances. The changes are minor and do not modify the overall behaviour observed.

Bloch-Johnson et al. (2021) study a multi-model set of idealized abrupt NxCO2 simulations. We first notice that a few individual models in Bloch-Johnson et al. (2021) go in the opposite direction, although only one model shows a negative sensitivity to both forcing and warming, which is consistent with our result. Also, the forcing of b990 and b025 (1.25x, 1.5x), which show the strongest non-linearity, is in a range which is not covered in Bloch-Johnson et al. (2021) and Meraner et al. (2013).
Apart from this, we are cautious against a direct comparison due to differences in our experimental setup, regarding both the initialization and the external forcing. Regarding the first point, our simulations start from different climate states (from historical and SSP5 scenario), rather than the pre-industrial climate. Regarding the forcing, our simulations also include other GHGs and aerosols, following historical+SSP5 forcing. If we expect the GHGs contribution to be in line with the CO2 one, aerosols might have a different impact, especially on the lower forcing cases. However, this alone is unlikely to explain the observed difference in the feedback parameter, since it would require an extremely large sensitivity of the feedbacks to the forcing agent, while most studies indicate at most a moderate sensitivity (Salvi et al. 2022, Richardson et al. 2019). For b990, if we consider an aerosol forcing around -1.0 W/m2 (with respect to a total of 1.8 W/m2 of instantaneous GHG forcing), the observed difference in the feedback parameter with respect to b100 would require an aerosol feedback of about -3 W/m2/K, which is not supported by current evidence.

We extended the discussion on this at lines 304-317. Further analysis is required to assess possible physical mechanisms, including exploring individual feedback contributions and performing a more quantitative estimate of the impact of other forcing agents, which we will leave to future studies.

2. I must say that I enjoyed reading the review comments from reviewer #2. She has raised many interesting and important research questions. I am wondering whether the authors could add one more discussion section, where you could discuss some unsolved questions or raise new open questions for further studies.

Following this suggestion, we briefly extended the last paragraph to present some pathways for future research at lines 547-556.

3. Another issue is whether the simulations reach a final equilibrium of the whole climate system. The authors have mentioned the "quasi-equilibrium" as defined by Li et al. (2013). I think the simulations reach the "quasi-equilibrium" state, where surface temperature is almost stabilized while TOA equals the deep-ocean heat uptake. I would suggest the authors move the reply to reviewer #2 about the "quasi-equilibrium" into the manuscript.

Thanks for the suggestion, we added a comment on the quasi-equilibrium at lines 246-250 of the revised version.

4. I would suggest the authors use different experiment names. The names of 'b990', 'b025',…,'b100' reflect the starting time of the simulations, but not the forcing difference. I would suggest the authors use an experiment name that could directly reflect the key information of the experiment, such as '1.25xCO2', '1.5xCO2','2xCO2',…,'4xCO2', or the direct CO2 concentrations.

The reviewer's suggestion would certainly help show the actual CO2 concentration of the runs, but we prefer leaving it as it is, because the starting year is a key information in our setup. In fact, in our setup also other GHGs and aerosol concentration change following the CMIP6 historical (b990) and SSP5-8.5 scenario (all other runs). The complete forcing set is then only completely defined through the starting year, which also gives information about the branching time (the simulations start directly from the historical/ssp585 simulations).